# A feasibility study to Reconstruct Atmospheric Rivers using spaceand ground-based GNSS observations

Endrit Shehaj<sup>1</sup>, Stephen Leroy<sup>2</sup>, Kerri Cahoy<sup>1</sup>, Juliana Chew<sup>1</sup>, Benedikt Soja<sup>3</sup>
<sup>1</sup>STAR lab, Department of Aeronautics and Astronautics, Massachusetts Institute of Technology, Cambridge, MA 02139, USA

<sup>2</sup>Atmospheric and Environmental Research, Lexington, MA 02421, USA

<sup>3</sup>Institute of Geodesy and Photogrammetry, ETH Zurich, Zurich, 8093 Zurich, Switzerland

Correspondence to: Endrit Shehaj (eshehaj@mit.edu)

#### Abstract.

Atmospheric rivers (ARs) are long filaments that transport large amounts of water vapor from the Tropics to mid- and high latitudes. They are directly related to heavy precipitation and extreme weather leading to flooding and mud slides. Accurate identification of AR structures over the ocean is important to improve the forecast of their landfall location and timing. Global Navigation Satellite Systems (GNSS) radio occultation (RO) is a space-based technique that can measure meteorological variables with high vertical resolution. While RO can observe structures like ARs in individual RO profiles, RO observations have non-uniform and sparse spatial and temporal sampling, so it is not yet possible to fully characterize AR morphology using RO alone.

In this work, we use previous research in which we applied machine learning (ML) to enhance the spatial and temporal resolution of RO observations. Here, we train neural networks (NNs) to map RO observations and help resolve ARs. Analyses using existing RO data, such as from the COSMIC-2 mission, showed that the sampling density is insufficient to resolve and geo-locate ARs. Adding observations from the other available missions (for example METOP) improved matters, but was still insufficient to reliably reconstruct AR structure.

We undertake a study to determine how many LEO RO satellites would be needed to quantify the structure, location, and timing of ARs. We simulate RO observations as would be obtained with Walker constellations of 12, 24, 36, 48 and 60 LEO RO satellites. First, we investigate possible constellations for proper AR monitoring. We aim for constellations that lead to hourly RO counts that change as little as possible during the AR (up to several days). This allows us to resolve ARs with similar accuracy during the scenario. We conclude that 3 or 6 orbital planes and inclinations between 85° and 90° perform best. Second, we make use of 12-h forecasts of the European Centre for Medium-range Weather Forecasts (ECMWF) system to interpolate the forecasts to the simulated RO constellation sampling coordinates. Third, we use the ECMWF-based RO observations to train ML models and map them to the ECMWF grid. We compare ML-mapped RO sampled grids to ECMWF products in a closed-loop validation. Initially, we map RO refractivity at 2 km geopotential height, where small-scale structures

related to water vapor are visible. We find that at least 36 RO satellites are needed to characterize the morphology of ARs in the Pacific basin with useful precision and accuracy (from the ML produced maps). Then, we use a framework with two consecutive NNs to map column-integrated water vapor (IWV) from profiles of RO. The first NN maps the refractivity into IWV, and the second NN maps the IWV spatially. In this case, we find that a constellation of 48 satellites is needed to continuously map IWV fields accurately and thus reconstruct the morphology of ARs with useful precision and accuracy. Finally, when using RO, we find that mapping refractivity into IWV is less accurate over land than over oceans. To further improve the AR mapping over land, we made use of IWV from ground-based (GB) GNSS. The significantly higher spatial and temporal resolutions of GB data compared to RO lead to much improved IWV fields and thus AR path and shape over land.

#### 40 1. Introduction


Global Navigation Satellite Systems (GNSS) radio occultation (RO) is a well-established remote sensing space technique, where GNSS signals are received by satellites in low-Earth orbit (LEO). The atmosphere along the signal path refractively bends the GNSS signals, and the induced delays can be converted into bending angles which can further be reduced to profiles of refractivity (Kursinski et al., 1997), (Kursinski et al., 2000), (Mannucci et al., 2021), (Melbourne, 2004). Using background atmospheric models, valuable information for temperature and water vapor is acquired by breaking down the refractivity values (Kursinski et al., 2000). The main features that make GNSS RO very attractive to meteorologists and climatologists are its long-term stability, all weather capabilities (not affected by clouds and rainfall), global coverage, absolute accuracy, high vertical resolution (100 m), and the fact that RO receivers are low-cost, low-power and compact sensors (Kursinski et al., 2000). The horizontal resolution of RO is 1.5 km in the cross-track direction. The horizontal resolution of an RO sounding in the along-track direction almost certainly depends on the effective vertical resolution of the retrieval. Through a 100-meter atmospheric layer, the horizontal path of an RO ray is ~70 km. This can be considered an optimistic horizontal resolution of an RO sounding.

In the horizontal dimension, RO data are heterogeneous in sampling density because of the uncoordinated orbital configuration of multiple RO spacecraft; GNSS constellations' orbits also lead to non-uniform RO observations. This leads to incomplete local time and meridional coverage as well as weak singularities at specific latitudes (Leroy et al., 2012). Additionally, RO sampling density has never been large enough to sample every cell of atmospheric synoptic variability, thus greater numbers of RO soundings should continue to improve our knowledge of the atmosphere without diminishing returns. (A cell is approximately described by the atmospheric Rossby radius of deformation (about 1000 km) and a span of several hours.)

To overcome the drawbacks of low horizontal sampling, (Leroy et al., 2012) used Bayesian inference to map RO data and study synoptic variabilities. (Shehaj et al., 2023) used neural networks (NNs) to further improve the horizontal mapping of RO data; additionally, NNs could significantly increase the temporal resolution. This is beneficial for weather phenomena

developing at short time scales. This research leverages the methodology developed in (Shehaj et al., 2023) using RO observations to resolve atmospheric rivers (ARs).





ARs are narrow maritime atmospheric low-level jets that transport large amounts of moisture from the Tropics into the mid- and high latitudes, often impinging on the continents (Newell et al., 1992), (Zhu and Newell, 1994), (Newell and Zhu, 1994). ARs can release massive amounts of moisture in the form of precipitation (NOAA, 2023). Depending on the size and intensity of the AR, it might lead to extreme precipitation (Leung and Qian, 2009), (Guan et al., 2010), (Lavers and Villarini, 2013), (Guan et al., 2016), (Lamjiri et al., 2017), (Chen et al., 2018), (Huning et al., 2019). ARs are related to higher risk of flooding events (Ralph et al., 2006), (Leung and Qian, 2009), (Lavers et al., 2011), (Konrad and Dettinger, 2017), (Curry et al., 2019), high sea water levels (Khouakhi and Villarini, 2016) and snow accumulation (Gorodetskaya et al., 2014). The amount of water ARs transport is comparable to average flow at the mouth of the Mississippi river (NOAA, 2023). In addition, their implicit connection to extratropical cyclone strength has been discussed (Zhu and Newell, 1994). (Zhang et al., 2018) show that ARs can contribute to the intensification of extratropical cyclones. The collection 'Atmospheric Rivers', a first effort containing selected research associated to ARs in *Geophysical Research Letters* (agupubs, 2019), encompasses many papers that show the connection of ARs to extreme weather events, the challenges/capabilities of weather prediction models to forecast/model/predict ARs, and the relationship between climate and ARs.

The more recent collection named 'Atmospheric Rivers: Intersection of Weather and Climate' in *Journal of Geophysical Research Atmospheres* (agupubs, 2024), presents further findings confirming the relationship between heavy precipitation/snowfall and ARs, capturing recent advances in numerical weather prediction (NWP) model capability to model/forecast ARs, comparing features in different ARs, evaluating the relationship between temperature and ARs, the effects of ARs on aerosols, modeling future ARs, and assessing the response of ARs to current climate change effects such as Arctic ice loss or mountain ice melting. (Zheng et al., 2021) shows that in-situ (dropsonde) observations along ARs improve forecast models and (Haase et al., 2021) shows that airborne ROs along rivers help distinguish key characteristics of ARs.

AR widths are typically 1000 km, and their lengths 2000 km or longer (Zhu and Newell, 1998). The total precipitable (column-integrated) water vapor is at least 20 mm (20 kg m<sup>-2</sup>) (Ralph et al., 2004). While they cover only 10% of Earth's circumference, ARs are still responsible for 90% of the total meridional moisture transport (Zhu and Newell, 1998).

Previous works have utilized RO for ARs. (Ma et al., 2011) shows the importance of assimilation of RO data in NWM for improving AR forecasting. (Bonafoni et al., 2019) review the importance of GNSS RO (and ground-based) data for observing, understanding and predicting extreme events. (Murphy and Haase, 2022) evaluate GNSS RO profiles in the vicinity of ARs. (Cao et al., 2024) and (Haase et al., 2021) study a campaign of airborne RO flying during AR events, describing the system, evaluating collected data, and showing their importance to distinguish AR characteristics, as well as the impact of the assimilation of airborne RO data in forecast models to predict ARs. (Rahimi and Foelsche, 2024) analyze specific humidity profiles and IWV from RO to study the vertical structure of ARs, concluding that ARs provide additional vertically-resolved information not contained in background or operational analyses.

Similar to GNSS RO, the high moisture in ARs is reflected in ground-based (GB) GNSS precipitable water vapor observations (Wang et al., 2019). While such data cannot help to reconstruct initial formation of ARs, the data are very useful to sense the intensity of ARs when they reach landfall. These observations are characterized by high temporal (up to few minutes) and horizontal (depending on the network) resolutions. Typically, GB GNSS networks are dense due to low costs and because they serve other purposes, for example monitoring of earthquakes (Blewitt et al., 2018).







Machine learning has been applied to RO data for different purposes, with many studies claiming promising results. (Hooda et al., 2023) uses machine learning to improve the water vapor retrieval from RO profiles. (Lasota, 2021) trains different machine learning models capable of retrieving tropospheric profiles of pressure, temperature and water vapor without using external data. (Chu et al., 2022) uses ML and RO to forecast wind fields. (Hammouti et al., 2024) and (Connor et al., 2021) apply ML to detect volcanic clouds and cloud signature, respectively. Other studies apply machine learning to ionospheric products of RO data; (Pham and Juang, 2015) aims to improve the retrieval of electron density for RO observations, and (Ji et al., 2024) detect ionosphere scintillation.

The goal of this paper is to utilize ML-mapped RO products to reconstruct the hourly evolution of ARs. The main questions that we address are: what type of RO constellation is necessary to resolve AR structures and, what is the optimal number of satellites that can accurately reconstruct our selected AR. We aim to resolve the total amount of water vapor present in ARs. In this context, RO profiles are mapped into hourly fields of IWV using two sequential NNs. Furthermore, we exploit the high temporal and spatial resolution IWVs from GB GNSS to further enhance the reconstruction of ARs over land. The RO signal dynamics become very complicated, and consequently tracking an RO signal in the planetary boundary layer (PBL) becomes difficult. Consequently, sometimes RO signals are not able to penetrate deeply into the PBL. In addition, present retrieval algorithms for RO must deal with several complications in the PBL such as spherical asymmetry (Ahmad and Tyler, 1998), (Ahmad and Tyler, 1999), RO signal loss, and truncation by operational retrieval systems. Also, RO retrievals are less precise due to apparent noisy behavior in the retrieved profiles of refractivity and water vapor. Most importantly, super-refraction induces negative biases in retrieved refractivity, which is associated with steep vertical gradients of water vapor and the resulting extreme bending of RO rays (Sokolovskiy, 2003), (Ao, 2007), (Xie et al., 2010). Advanced algorithms have been proposed to ameliorate such biases in retrieval (Xie et al., 2006), (Wang et al., 2020), (Wang et al., 2024) but have yet to be applied to program-of-record data. Before applying the ML algorithms above to actual RO data to measure AR water vapor content, RO retrieval algorithms that account for super-refraction must be applied operationally. Nevertheless, because the goal of our project is one of RO sampling densities, we assume perfect precision and accuracy in RO retrieval, leaving improvement in retrieval performance in the PBL to other ongoing research programs. In addition, it is important to point out that often RO data can be exploited down to the surface right in center of the ARs, where WV values are high, but gradients are not.

Section 2 describes the selected AR scenario, the methodology and initial results using current RO data. Section 3 shows the chosen LEO constellations and the simulated ROs data using European Center for Medium-Range Weather Forecasts (ECMWF) model; then, it displays the results of different LEO constellations to reconstruct ARs. Section 4 shows the results

when using currently available actual RO observations. Section 5 shows the enhancement of AR mapping over lands using GB GNSS. Section 6 summarizes the results and future work.

# 2. AR scenario and method to map RO data

In this section, we present the AR scenario that we use for our analysis and give an overview of the ML method used to map the RO observations to reconstruct the AR.

# 2.1 AR scenario





We focus on the North Pacific and ARs that landfall along the west coast of North America. In this region, ARs are mainly responsible for extreme precipitation events, contributing to 30–50% of the annual precipitation (Dettinger, 2013). *Figure 1* depicts the AR scenario reported on the website of US National Weather Service, (NOAA, 2021). The left plot of *Figure 1* shows the region (90°W–160°W and 10°N–60°N), and the right plot of *Figure 1* depicts the AR as a blue stream from the Pacific Ocean towards the US coast. The AR was visualized using ECMWF forecast data.

Figure 1: AR scenario visualized using 12-h forecast ECMWF data, with a resolution of 0.1°. The left panel shows the scene, and the right panel shows the refractivity at 2 km height on the 24<sup>th</sup> of October 2021 at 19:00 UTC, (similar to (Shehaj, 2023)). The purple rectangle highlights the AR.

This AR lasted about 2 days during 24th and 25th of October 2021. It was indicated in long range models by October 18<sup>th</sup>, and on October 20<sup>th</sup> the models were showing a high to moderate AR that could lead to high precipitation. This was abnormal considering the AR was happening early for the region (NOAA, 2021). The river evolved together with a bomb-cyclogenesis cyclone which developed above the river. Heavy rainfall precipitated on October 24<sup>th</sup> in the San Francisco Bay Area of the U.S. The AR led to strong winds, flood warnings in all the region, storms that caused flooding, fallen trees, power outages and minor mudslides. The event also generated large and powerful waves along the Pacific coast of the U.S. October 24<sup>th</sup> was the wettest day ever for many cities around the San Francisco Bay Area, as reported in table 'Heavy Rain' in (NOAA, 2021).

The evolution of the selected AR can be seen in the supplementary material where videos of ECMWF refractivity and IWV fields are included.

# 155 **2.2 Machine learning**





The machine learning (ML) approach used here is based on (Shehaj et al., 2023), in which we mapped global climatologies. In this work, we focus on mapping RO data at a more regional scale where the AR occurs. Our ML algorithm is a classical artificial fully connected NN, where the first layer has the inputs, the last layer the outputs and each neuron of one layer is connected to all the neurons of the previous layer (Haykin, 2009). This is also known as Multilayer Perceptron (MLP). NNs are one of the most important algorithms in ML, proven generalizable across several fields. We have successfully used NNs in previous research to map RO or GB GNSS tropospheric products (Shehaj et al., 2023), (Shehaj, 2023), (Shehaj et al., 2023), (Miotti et al., 2020).

The loss function that the network aims to minimize is the mean squared error (mse) between the targets and model output, the latter of which is dependent on the weights and biases of each neuron. The stochastic gradient descent is used to find the local minimum of the function. The nonlinearities between inputs and outputs are defined using the activation function, in this case the Rectified Linear unit Function (ReLu) (Nwankpa et al., 2018). To avoid numerical issues, we have standardized the feature data before training, centering them around zero and using the variance to scale them, as typically performed in many ML applications.

After the overview of the scenario and the ML method, the next section presents our approach to resolve ARs.

# 170 3. Approach to detect ARs from RO and ML

In this section, we present our approach to reconstruct ARs using simulated ROs in a closed-loop validation. In a first step, we investigate different LEO constellations' designs to define suitable constellations for monitoring ARs. We use the LEO and GNSS orbits to define the locations and times of ROs, and the ECMWF weather model to simulate RO observations. In a second step, we train ML based on the simulated RO observations, and then map the RO quantities to the ECMWF grids for a closed-loop validation. We evaluate our approach for RO-based refractivities and IWV, by analysing the reconstruction of the AR for different LEO constellations to conclude on the necessary minimum number of satellites.

#### 3.1 First step: Simulated RO observations of LEO constellations designed for AR detection

We investigate the sounding density that would be required to analyse the location and morphology of an AR. Commercial RO companies around the world are now projecting RO sounding densities exceeding 100,000 daily, (https://spire.com/blog/maritime/radio-occultation-in-less-than-500-words/, 2021). For this purpose, we will examine different LEO constellation designs in order to conclude a) what constellations are suitable to continuously and accurately monitor ARs,

and b) what is the minimum number of satellites that leads to an appropriate amount of RO observations for our purpose. Here, we define the amount of RO observations as appropriate if it is sufficient to reconstruct the humidity in the AR. We choose to work with Walker constellations because they promise the most uniform possible RO coverage globally and in local time and because they are infinitely scalable. Examples of constellations designed for uniform coverage are the medium Earth orbit (MEO) GNSS constellations themselves, which were developed to meet specifications of the minimum number of satellites that would be visible above the horizon at any time from any point on Earth. Although we will focus our analysis on the AR scenario located in the US West Coast, discussed in Section 2.1, we expect our results to be valid for ARs occurring in other regions globally.

In Section 3.1.1, we analyse different orbital parameters to design an optimal constellation for monitoring ARs. In Section 3.1.2, for the defined constellations, we display simulated RO refractivity and IWV based on NWM data.

# 3.1.1 Simulated Walker Constellations






We test Walker constellations with 12, 24, 36, 48, and 60 satellites, to decide on the minimum number of satellites appropriate for monitoring ARs. For all constellations, we set the eccentricity equal to 0 and the altitude to 800 km. Every LEO satellite can track RO events of 30 Global Positioning System satellites, 23 GLONASS satellites, 18 Galileo satellites, and 49 BeiDou satellites. These satellites were all operational on the dates of the AR event described in Section 2.1. The GNSS satellite orbits are taken from the two-line element records of Celestrak (https://celestrak.org/NORAD/elements/, 2024) for October 2021 and are propagated in time using SGP4 (https://pypi.org/project/sgp4/, 2024). To evaluate constellation performance, we count the number of RO observations in the AR region that results from using different Walker constellation configurations. We also consider the temporal uniformity of the sampling density since we aim to monitor and reconstruct the AR structure continuously in time.

The free parameters of a Walker constellation are the number of orbit planes uniformly distributed in right ascension of the ascending node (RAAN), the number of satellites in each orbit plane distributed evenly in argument of latitude, the eccentricity and argument of perigee of each plane, the altitude of the orbit planes, and a parameter describing how the satellites' orbit anomalies (timing) are staggered between planes. We consider only circular orbits, so the eccentricity is zero and the argument of perigee is undefined. We also fix the altitude of the orbits to 800 km. We compute the hourly RO counts for inclinations every 5° in the interval 65° to 100°, with 3–6 orbit planes, and with 12, 24, 36, 48, and 60 total satellites. Over four-days, an inclination of 100° corresponds to a nearly sun-synchronous orbit.

Figure 2 displays the hourly number of RO counts in the AR region (10°N to 60°N and 90°W to 160°W) for an ensemble of the RO count simulations. Figure 3 displays the hourly RO counts in the AR region for inclinations of 85° for 3, 4, and 6 orbital planes for constellations of 12, 24, 36, 48, and 60 satellites. Table 1 summarizes the statistics in terms of average and standard deviation (SD) of the hourly RO counts. The Walker constellation parameters that exert the greatest influence on the hourly number of RO observations in the AR region are the number of orbit planes and their inclination for a fixed number of

satellites. The inclination changes the distribution of RO soundings since low inclination leads to absence of coverage at high latitudes. For example, the COSMIC-2 mission with an inclination of 24° supports collecting RO up to ~45° latitude.

This analysis has several prominent outcomes. (1) Three- and four-plane configurations show strong periodicity in the hourly RO counts. The periodicity itself depends on the number of orbital planes and on their inclination, and the timing of the extrema depends on inclination. (2) The least temporal variation of hourly RO counts for 3 planes happens at 90° inclinations, and for 4 planes at 65° inclination. For 6 planes, the temporal variation of hourly RO counts is very small for all inclinations, with a minimum at 85° inclination. (3) For constellations with 3 planes, inclinations of 65° and 100° lead to the largest average hourly RO counts. 85° inclination leads to the least average hourly RO counts; see Table 1. For constellations with 3 planes, inclinations of 65° lead to the highest variation of hourly RO counts, while 90° inclinations lead to the lowest variation. (4) For constellations with 4 planes, inclinations of 65° lead to the largest average hourly RO counts, while 95° inclinations lead to the lowest average hourly RO counts. For constellations with 4 planes, inclinations of 95° lead to the largest variation of hourly RO counts. (5) For constellations with 6 planes, inclinations of 65° lead to the largest average hourly RO counts. For constellations with 6 planes, inclinations with 6 planes, inclinations of 65° lead to the largest variation of hourly RO counts, while 85° inclinations (70° for 12 satellites) lead to the least variation of hourly RO counts.

Heuristically, any LEO RO receiver in a high-inclination orbit obtains approximately 500 soundings per day per tracked GNSS constellation. This derives from a single satellite orbiting the Earth approximately 14 times daily and there being approximately 30 transmitters in a GNSS constellation, with the RO receiver tracking both rising (fore-viewed) RO soundings and setting (aft-viewed) RO soundings.

Figure 2: Hourly number of RO counts for Walker constellations of RO receivers in orbit planes with inclinations of 65°, 80°, 90°, and 100° in the AR region. On the left are displayed the cases for 12-satellite constellations and on the right the cases for 60-satellite constellations. The top, middle and bottom panels represent the cases of 3, 4 and 6 orbital planes.

Figure 3: Hourly number of RO counts for 85°-inclination orbit planes, in the AR region, for 12, 24, 36, 48, and 60 satellites. The top, middle and bottom panels represent the cases of 3, 4 and 6 orbital planes.

Table 1 Temporal means and standard deviations (SD) of hourly RO counts for constellations of 12 and 60 satellites, for 3, 4 and 6 planes, and for inclinations ranging from 65° to 100°.

| 12 satellites   |          |       |       |       |       |       |       |       |       |
|-----------------|----------|-------|-------|-------|-------|-------|-------|-------|-------|
| Inclina         | tion [°] | 65    | 70    | 75    | 80    | 85    | 90    | 95    | 100   |
| 3               | Mean     | 99.7  | 95.4  | 96.3  | 95.9  | 90.8  | 93.7  | 93.2  | 99.3  |
| planes          | SD       | 32.9  | 23.0  | 23.5  | 17.0  | 9.0   | 6.6   | 11.5  | 20.2  |
| 4               | Mean     | 98.5  | 98.1  | 96.9  | 94.2  | 93.3  | 92.4  | 91.9  | 98.2  |
| planes          | SD       | 10.5  | 12.0  | 14.6  | 16.8  | 17.7  | 16.7  | 18.6  | 15.0  |
| 6               | Mean     | 98.5  | 98.1  | 95.8  | 94.4  | 91.6  | 92.7  | 94.7  | 97.5  |
| planes          | SD       | 8.0   | 7.1   | 7.2   | 7.5   | 7.2   | 7.9   | 7.2   | 7.6   |
| 60 satellites   |          |       |       |       |       |       |       |       |       |
| Inclination [°] |          | 65    | 70    | 75    | 80    | 85    | 90    | 95    | 100   |
| 3               | Mean     | 496.7 | 478.4 | 482.9 | 479.8 | 452.9 | 468.5 | 466.5 | 497.5 |
| planes          | SD       | 164.0 | 114.4 | 116.4 | 82.6  | 40.7  | 28.7  | 55.5  | 98.8  |
| 4               | Mean     | 492.1 | 490.1 | 483.3 | 471.7 | 465.3 | 462.8 | 459.3 | 489.6 |
| planes          | SD       | 45.3  | 50.3  | 68.1  | 79.1  | 81.8  | 76.6  | 88.0  | 69.2  |
| 6               | Mean     | 492.8 | 491.5 | 481.7 | 472.9 | 458.4 | 463.2 | 471.9 | 490.6 |
| planes          | SD       | 32.2  | 25.0  | 23.2  | 25.0  | 20.3  | 28.9  | 28.2  | 25.5  |

Using larger numbers of RO soundings and temporal uniformity as the criteria, we settle on the final configurations for our ML exercises:

- 12 satellites: 3 planes and 90° inclination, yielding 145K global soundings, 8,992 in the AR region;
- 24 satellites: 6 planes and 85° inclination, yielding 290K global soundings, 17,571 in the AR region;
- satellites: 6 planes and 85° inclination, yielding 434K global soundings, 26,343 in the AR region;
- satellites: 6 planes and 85° inclination, yielding 579K global soundings, 35,153 in the AR region;
- satellites: 6 planes and 85° inclination, yielding 724K global soundings, 43,956 in the AR region.

Table 2: Final configurations of the LEO constellations.

| Number of LEO satellites | Planes | Inclination<br>[°] | Number of global soundings [K] | Number of soundings in the AR region |
|--------------------------|--------|--------------------|--------------------------------|--------------------------------------|
| 12                       | 3      | 90                 | 145                            | 8,992                                |
| 24                       | 6      | 85                 | 290                            | 17,571                               |
| 36                       | 6      | 85                 | 434                            | 26,343                               |
| 48                       | 6      | 85                 | 579                            | 35,153                               |
| 60                       | 6      | 85                 | 724                            | 43,956                               |

The RO counts in

Table 2 are for the entire 4-day period of 23–26 October 2021.

From our simulations, we achieve about 180K occultations daily with a constellation of 60 satellites. While the number of current LEO RO satellites is already around 60 satellites, the number of RO counts reported is much lower compared to our simulation. This is because of different factors, for example, some LEO RO constellations do not collect data from all available

GNSS, in real constellations some data are not reported or difficult to obtain statistics on because they are considered 'bad' or poor-quality observations. In addition, different LEO RO satellites only collect data up to a certain latitude, e.g., COMSIC-2 only collects data from 45°S to 45°N. Finally, as we will see in the following sections, the goal is to simulate a high-performing constellation in order to reconstruct the IWV field in the AR scene; we find that the statistical improvement is relatively small when we consider 60 satellites compared to 48 satellites (see Section 3.2.2 and Section 3.2.3.2), so we did not expand the constellation beyond 60 in this work.

We also studied constellation design for another AR in the UK region between September 30<sup>th</sup> and October 3<sup>rd</sup>. The hourly number of ROs and the number of planes is similar to the event on the US West Coast. Differences appear for the inclination, where for the UK event an inclination of 100° appears more appropriate. This difference is mainly caused by the different GNSS TLEs for the two events. Using the same TLEs leads to similar results for both scenarios. We point out that for 6 planes, the variation of hourly RO counts is similar for the different inclinations. The variations for different inclinations for 3 and 4 planes are much more noticeable, as also reported in Table 1.

# 3.1.2 Refractivity and IWV from ECMWF






In this section, we display the simulated RO observations that result from the selected Walker constellations in Section 3.1.1. We use 12- to 23-hour forecast fields (at an hourly cadence) from the ECMWF operational 4DVar data assimilation system, and the time and location where ROs geometrically occur, to simulate the RO observations. Fields are published on  $0.1^{\circ}$  grid in latitude and longitude. We used pressure p, temperature T, and water vapor pressure  $p_w$  to compute refractivity (Rueger, 2002):

$$N = (n-1) \times 10^6 = (77.6890 \text{ K hPa}^{-1}) \frac{(p-p_w)}{T} + (71.2952 \text{ K hPa}^{-1}) \frac{p_w}{T} + (375463 \text{ K}^2 \text{ hPa}^{-1}) \frac{p_w}{T^2}, \tag{1}$$

When interpolating the model to the times and locations of RO, we took the model refractivity profile in the cell nearest to the RO sounding and interpolated linearly the vertical dimension. *Figure 4* displays the simulated refractivity at 2 km geopotential height. The refractivity plots show structures related to water vapor, which are also strongly correlated with boundary layer clouds. The AR structure is not obvious in these plots that collect four days of simulated refractivity because the AR changes position in that time frame.

Figure 4: Simulated refractivity at 2 km geopotential height, based on ECMWF, for constellations of 12, 24, 36, 48, and 60 satellite constellations, during 23 to 26 October 2021.

Figure 5 displays the simulated refractivity based on ECMWF, for the case of 12 satellites LEO constellation, for 1 km to 10 km geopotential heights above the ground. The refractivity at different heights above ground shows similar structure over land. This reflects the terrain altitude (not shown here). We will use the refractivity at different altitudes above ground to map the IWV.





Figure 5: Simulated refractivity from 1 to 10 km height above the ground, based on ECMWF, during 23 to 26 October 2021. Here we display the case of 12-satellite constellations.

We use an ML approach to retrieving column-integrated water vapor given refractivity values at discrete geopotential heights at 1-km intervals above the surface topography up to 10 km above the surface. ECMWF forecasts are used as the training data set. ECMWF is a layer-based model that publishes temperature and humidity intended to represent a layer of fluid and the pressures and heights at the interfaces between the model layers (ECMWF, 2024). Layer indices are denoted as integers and layer boundaries as half-integers. The pressure for layer i is determined by  $p_i = (p_{i-1/2} + p_{i+1/2})/2$  and similarly for layer heights  $h_i$ . The integrated water vapor (IWV) for layer i is calculated by

$$IWV_i = Q_i \left( p_{i+1/2} - p_{i-1/2} \right) / g_0 \tag{2}$$

where,  $Q_i$  is the specific humidity for layer i,  $g_0$  is the WMO standard mean sea-level acceleration due to gravity, equal to 9.80665 m s<sup>-2</sup>. The total column IWV is the sum of the individual layer integrated water vapors over all model layers. The ECMWF operational data assimilation system has 137 model layers.

Figure 6 displays IWV calculated from ECMWF at the simulated RO geolocations. The IWV (from the ground) is a useful meteorological observation to model the total amount of water vapor in the AR structure. Again, from these stacked datasets (over 4 days), the AR shape and path is not obvious.



Figure 6: IWV, based on ECMWF, for constellations of 12, 24, 36, 48, and 60 satellites, during 23 to 26 October 2021.

# 3.2 Second step: Machine learning framework to detect AR from RO

This section displays the results of ML applied to ECMWF-based RO simulated data to reconstruct the AR structure for the constellations simulated in Section 3.1. Section 3.2.1 summarizes the input variables, output variables and hyperparameters for the different NNs used in this work. Section 3.2.2 shows the results of reconstructing the AR at 2 km height. In Section 3.2.3, we show the results of reconstructing the entire IWV in the AR scenario, using two consecutive NNs. The first NN is used to map refractivity profiles into IWV (called mappable-IWV) and the second NN is used to spatially interpolate the mappable-IWVs.

In this work, we aim to use RO to reconstruct (and monitor) an AR. By reconstruction we consider the capability of the ML model to produce fields of refractivity and IWV that describe the spatial and temporal morphology of ARs and quantify moisture associated with them to a degree sufficient for atmospheric studies. The reconstructed fields can be used to continuously monitor ARs.

We also point out that the ML-mapped quantities (such as refractivity and IWV) are the result of an ensemble of 10 different NN trained models. This makes our results more general and robust to randomness of the trained model caused by initialization of model parameters, stochastic optimization algorithms that randomly sample the data points or possible GPU precision and optimization implementation (Altarabichi et al., 2024).

#### 3.2.1 Architecture and hyperparameters tuning

In this work, we test two approaches to reconstructing the AR structure. In the first approach, we train NNs that can map the refractivity at 2 km iso-height with high horizontal and temporal resolution. The NN-mapped fields displayed here have a temporal resolution of 1 hour and a horizontal resolution of  $0.5^{\circ}$ ; these resolutions are enough for our visualization/evaluation; however, it is possible to interpolate at a higher resolution. These numbers are sufficient to resolve the spatial and temporal scales of ARs (Zhu and Newell, 1998). The NN learns to map the refractivity from the geolocations (and times) of the RO soundings. In the second approach, we used refractivity profiles in the geopotential height interval 1–10 km above the ground in 1-km intervals to compute and map IWV with high horizontal and temporal resolution. In this case two sequential NNs are used. The first NN maps refractivity profiles to IWV for each refractivity profile. In this step, the surface geopotential height

and sine and cosine of local (solar) time are also input to the NN. The second NN is trained to spatially (and temporally) map the IWVs from the soundings horizontally, including geolocation, surface geopotential, and UTC time as inputs.

Table 3 provides a summary of all the NNs used in this work. In addition to the input and target variables, we also show the chosen hyperparameters and architectures of the NNs, which are selected based on the statistics of the validation set. These parameters are selected after tuning each of the NNs individually for the three different mappings—spatial mapping of refractivity, computing IWV from refractivity profiles, and spatially mapping IWV— for the different satellite constellations (12, 24, 36, 48, 60). We formulate a NN with standard hyperparameters for each of the three NN types; see Table 3. When mapping refractivity, we consider 30,000 epochs and a batch size of 100; however, the learning rate and the number of layers depend on the number of satellites in the Walker constellation. When computing IWV from refractivity profiles, we use 30,000 epochs, a batch size of 50, a learning rate of  $1 \times 10^{-4}$ , and always 5 layers. When computing maps of IWV from IWV soundings, we use 30,000 epochs, a batch size of 100, a learning rate of  $1 \times 10^{-3}$ , and 5 layers. Considering that the postfit residuals can vary slightly when applying different models, different ML algorithms, or a (slightly) different input dataset, further tuning is not necessary.

Table 3 Summary of the NNs developed for this work. Here, we show the input and target variables, as well as the tuned hyperparameters and architecture of the NNs. Only the hyperparameters whose tuning affects the results are displayed here.

|                                            | Input                                                                                        | Output | Constellation | Hyperparameters |               |            | Architecture     |                                |
|--------------------------------------------|----------------------------------------------------------------------------------------------|--------|---------------|-----------------|---------------|------------|------------------|--------------------------------|
|                                            | _                                                                                            | _      | ·             | Epochs          | Learning rate | Batch size | Number of layers | Number of neurons              |
| Mapping of                                 | - Latitude                                                                                   | N      | 12 satellites |                 | 1e-3          |            | 7                |                                |
| refractivity                               | - Longitude                                                                                  |        | 24 satellites |                 | 1e-4          | -          | 10               | =                              |
| at 2 km                                    | - UTC time                                                                                   |        | 36 satellites | 30'000          | 1e-4          | 100        | 10               | _'                             |
|                                            |                                                                                              |        | 48 satellites |                 | 5e-5          | _'         | 10               | _'                             |
|                                            |                                                                                              |        | 60 satellites |                 | 5e-5          | ='         | 10               | =                              |
| Refractivity<br>1-10 km<br>above<br>ground | - N 1 km<br>- N 2 km<br><br>- N 10 km                                                        | IWV    | All cases     | 30,000          | 1e-4          | 50         | 5                | Layer 1: 512<br>Layer 2:n: 128 |
| mapped to IWV                              | <ul><li>Cosine of LT</li><li>Sine of LT</li><li>Surface topography</li></ul>                 |        |               |                 |               |            |                  |                                |
| Mapping of ground IWV                      | <ul><li>Latitude</li><li>Longitude</li><li>UTC time</li><li>Surface<br/>topography</li></ul> | IWV    | All cases     | 30,000          | 1e-3          | 100        | 5                | -                              |

# 3.2.2 Horizontal mapping of refractivity at 2 km iso-height




We trained ML using the simulated refractivity data at 2 km shown in *Figure 4*, in Section 3.1.2. Initially, we randomly split the data into training and test datasets, where 80% was used for training and 20% for testing. We used a random 10% subset of the training dataset for validation. We computed the SD and mean relative error (MRE) of the residuals of the test

dataset for the constellations of 12, 24, 36, 48, and 60 satellites. The post-fit residuals had SDs of 6.1, 5.5, 4.7, 4.6, and 4.3 *N*-units and mean-residual-errors of 1.6%, 1.4%, 1.2%, 1.2%, and 1.1% for those same 5 constellations in the given order. Post-fit residuals are reduced when using more satellites no matter what metric we use; nonetheless, diminishing returns on the increase in the number of satellites becomes noticeable at 36 satellites.




We validate our approach by comparing the ECMWF forecasts to our ML analysis interpolated onto the ECMWF longitude-latitude grid. *Figure* 7 (top panels) displays ECMWF and the ML analyses of the refractivity field at 2 km geopotential height for one epoch during the AR and the differences. The reconstructed field improves with the increasing number of satellites. With 12 and 24 satellites, an important part of the river is not reconstructed well. We need at least 36 satellites to reconstruct the AR well and we need 60 satellites to also model some parts of the cyclone off the shore of British Columbia. The presence of the AR is apparent with all five constellation configurations; increasing the number of satellites, however, has the effect of refining the horizontal resolution and fine-scale structures in the AR. The improvement in fine-scale structure is reflected in the smaller residuals with increasing number of satellites, shown in *Figure* 7 (bottom panels). The improved horizontal resolution is further confirmed by the smaller SDs of the differences between ECMWF and ML-mapped grids for increasing number of satellites, shown in *Figure* 8. Increased horizontal resolution also leads to better estimations of maxima in refractivity associated with AR along its entire length. Similar figures are produced for the entire scenario and stacked together as a movie, attached to this publication (see supplement material of this paper).

Figure 7: Refractivity fields at 2 km geopotential height, for 25 October 2021, at 03:00. Top panels: the left panel shows the ECMWF forecast field and the other panels show the ML-based fields when using constellations of 60, 48, 36, 24, and 12 satellites. Bottom panels: differences between the ECMWF field and the ML mapped fields.

The hourly time series of the SD of the residuals (in the bottom panels of *Figure 7*) are displayed in *Figure 8*. We can see the improvement from 12 to 24 satellites and a clear improvement from 24 to 60 satellites. The average SD for the 4 days duration of the scenario are 6.6, 5.7, 5.1, 4.8, 4.6 *N*-units for 12, 24, 36, 48, and 60 satellites, respectively.

Figure 8: Time series of the SDs of the hourly differences between the ECMWF refractivity field and the ML-mapped refractivity fields at 2 km geopotential height for the different satellite constellations.

# 3.2.3 AR from IWV framework


We use a framework with 2 sequential NNs to reconstruct the AR shape in terms of IWV using RO profiles and to perform a closed-loop validation with the ECMWF grid. *Figure 9* displays the flow chart, composed of three blocks: (1) we simulate the observations based on ECMWF, (2) we infer IWV from refractivity profiles, and (3) we map IWV in the horizontal. In this way we produce continuous IWV fields and compare them with the original ECMWF IWV analyses.

Figure 9: Flow chart of mapping IWV fields from simulated RO profiles. The flow chart consists of three main parts. The first part is the simulation of the observations where the orbits of the LEO Walker constellations (and GNSS), and ECMWF forecast grids are used to simulate RO observations. The second part is the first NN where refractivity profiles are mapped into IWV over the surface topography. The sine and cosine of the local time and the surface topography are also used as input variables. The output of this part are the mappable-IWVs. The third part is the second NN where model is trained to map the mappable-IWVs spatially. This model can be used to produce grids of IWVs (same locations as ECMWF grid) and compare the ML-mapped IWV fields with ECMWF fields, in a closed-loop validation.


We include all RO locations simulated using the propagators. Simulation studies have shown that only 70% of potential RO soundings are actually recorded by actual RO missions. We point out that to make our experiments as realistic as possible, as shown in the flow chart, only 50% of the simulated RO profiles are used to train the 1<sup>st</sup> NN. The 2<sup>nd</sup> 50% is used to produce IWVs at untrained locations and they will train the 2<sup>nd</sup> NN to map the final IWV fields and study the AR structure. Using 100% of the data to train the 1<sup>st</sup> NN is not realistic because the resulting mappable-IWVs are not independent. Indeed, in a real-case scenario, we would obtain refractivity profiles from RO, and to map the profiles into IWVs we must use already trained NN models. Clearly these NN models must be trained using other datasets.

Considering that in RO processing we lose many observations because of processing adversities and not being able to process all current GNSS satellites, this amount of data can be considered conservative.

# 3.2.3.1 IWV inferred from refractivity profiles

We trained NNs for the different constellations where we mapped the refractivity from 1 km to 10 km height above the ground to IWV. We use refractivity at 1-km intervals up to 10 km above the surface for several reasons. The choice of profiles in the 1-10 km interval is conservative considering that in the lowest part of the atmosphere RO-retrieved refractivity is less accurate due to ducting and super-refraction, multipath, SNR attenuation or spherical symmetry in the atmosphere which have less accuracy for strong horizontal gradients. There is no simple retrieval of water vapor from refractivity values in the lower atmosphere without a prior, such as an atmospheric forecast, thus preventing retrieval and vertical integration of water vapor to obtain a column-integrated value. Thus, we rely instead on a few indicative values in the PBL and information on the atmospheric dynamical state, which information is contained in values throughout the tropospheric column. A good machine learning algorithm should be able to discern the meteorology of the local environment and draw on this information to establish a relationship between the refractivity values in the lower troposphere, where water vapor contributes most to refractivity, and column-integrated water vapor. Our motivation is justified by our results, which are found in Table 4.

The results in terms of residuals on the 50% dataset not used for training are displayed in *Figure 10*. For simplicity, we display only 3 constellations. For constellations with 12, 24, 36, 48, and 60 satellites, respectively, the SDs of the residuals are 2.0, 1.7, 1.6, 1.5 and 1.5 mm of precipitable water, while the MREs are 8.4%, 6.8%, 6.2%, 5.7% and 5.5%. Note that 1 mm of precipitable water is equal to 1 kg m<sup>-2</sup> of mass per unit area of column IWV. These statistics reflect the importance of larger amounts of data to map IWV. In addition, *Figure 10* shows regions of higher than usual residuals. One such region is the ocean-continent boundary, caused by micrometeorological phenomena associated with those boundaries. Another region is the Colorado Plateau, where most of the water vapor is trapped in the lowest 1 km part of the atmosphere because of large-scale subsidence. Inferred IWV in this circumstance is expected to be erroneous because our NN uses no input below 1 km height above the surface. In addition, errors in IWV over the continents are enhanced because of micrometeorology. Our NN trains only on a subset of RO soundings, which cannot be completely representative of all micrometeorological environments because of small-scale spatial heterogeneity, leaving some micrometeorological environments unsampled and thus subject to erroneous inference by the NN. Lastly, large residuals seen over the ocean are caused by abrupt horizontal discontinuities in IWV, especially near the AR itself, where large refractivity gradients occur within a few kilometers.

Figure 10: Differences between ML-mapped IWVs from refractivity profiles and IWVs simulated from ECMWF. The mapped IWVs are produced for the 50% of the dataset that was not used to train the ML model. Here we display the differences for constellations of 12, 36 and 60 satellites.

In order to account for micrometeorological influences of topography we including topography as defined by the ECMWF model as input parameter for the NN for each sounding. This is to account for the fact that the terrain on continents varies much more significantly than the terrain over ocean. In addition, we added the sine and cosine of solar angle (local time) as input variables in order to provide the neural network additional information for the on-shore/off-shore flow and related diurnal cycles of water vapor. Both the sine and the cosine are provided in order to circumvent discontinuities across midnight that arise when instead providing a scalar on a finite [0,24] hour interval.

Using only 50% of the data for training the model, we lose an important part of the observations. We also performed tests where we used 80% of the entire dataset to train the model and 20% to test it. The results for the test dataset are summarized in Table 4. We also computed separate statistics of the mapped IWVs over ocean and over land. The statistics over land and over ocean are similar in terms of SD, however MRE reveals elevated error over land relative to over ocean. We also trained models to learn the mapping over ocean and over land independently. We found no notable improvement in performance in the latter experiment; thus, we proceed to only consider joint land-ocean learning only. The results of mapping separately ocean and land show that the IWVs over ground, visible in Figure 6, are more difficult to map. One reason is the larger variation of the surface topography. For this reason, in Section 5, we augment the IWV data set with ground-based GPS station measurements of IWV.

Table 4 Statistics of the differences between the ML mapped IWVs from refractivity profiles and the IWVs simulated using ECMWF model, for the 80% test dataset. The statistics, in terms of SD and MRE, are computed for the entire scene, for only the IWVs over ocean and for the IWVs over land.

|            | N unit  | 12 satellites | 24 satellites | 36 satellites | 48 satellites | 60 satellites |
|------------|---------|---------------|---------------|---------------|---------------|---------------|
| All data   | SD [mm] | 1.8           | 1.6           | 1.4           | 1.4           | 1.3           |
|            | MRE [%] | 7.6           | 6.0           | 5.5           | 5.2           | 4.8           |
| Over Ocean | SD [mm] | 1.6           | 1.6           | 1.4           | 1.4           | 1.3           |
|            | MRE [%] | 6.0           | 5.3           | 4.4           | 4.4           | 4.1           |
| Over Land  | SD [mm] | 1.8           | 1.5           | 1.4           | 1.3           | 1.3           |
|            | MRE [%] | 9.7           | 7.3           | 6.8           | 6.4           | 6.2           |

# 460 3.2.3.2 Continuous IWV fields from mappable-IWV

We use the mappable-IWVs produced from the 1<sup>st</sup> NN (the NN that maps refractivity into IWV) to map continuous fields of IWVs. The error of the mappable-IWVs was shown in Section 3.2.3.1 (for example Figure 10 shows their differences to ECMWF IWVs). The mappable-IWVs, used to map IWV spatially (and thus produce continuous IWV fields), are for the profiles not trained in the 1<sup>st</sup> NN. This is for 50% of the simulated ROs, as shown in the flow chart in *Figure 9*.

Figure 11 (top panels) displays the IWV fields, for one epoch during the AR, for the ECMWF forecast model (left panel) and the reconstructed fields mapped with ML, for the different constellation configurations (60, 48, 36, 24, and 12 satellites from left to right). The bottom row displays the difference between the ECMWF field and the ML-mapped fields. The reconstructed field clearly improves with increasing number of satellites. From the residuals (bottom panels of Figure 11), we also see how the number of large value residuals (dark blue or dark red) decreases with increasing number of satellites. Again, similar figures are produced for the entire scenario and stacked together as an animation, available as supplementary material.

Figure 11: IWV fields, for 25 October 2021, at 03:00. Top panels: the left panel shows the ECMWF forecast field and the other panels show the ML-based IWV fields when using constellations of 60, 48, 36, 24 and 12 satellites. Bottom panels: differences between the ECMWF field and the ML mapped fields.

The hourly SD of the residuals (bottom panels of *Figure 11*) are displayed in *Figure 12*. Both the SD and the MRE are reduced with increasing numbers of satellites in the constellation, but improvement becomes marginal beyond 48 satellites. For 12, 24, 36, 48 and 60 satellites the average SDs of the differences are. 3.2, 2.6, 2.2, 2.1 and 2.0 mm; an increment of 12 satellites improves the results by  $\sim$ 19% (12 to 24 satellites),  $\sim$ 15% (24 to 36 satellites),  $\sim$ 5% (36 to 48 satellites) and  $\sim$ 5% (48 to 60 satellites). The average MREs (not visualized here) are 12.1%, 9.9%, 8.7%, 8.0% and 7.5%; an increment of 12 satellites improves the results by  $\sim$ 18% (12 to 24 satellites),  $\sim$ 12% (24 to 36 satellites),  $\sim$ 8% (36 to 48 satellites) and  $\sim$ 6% (48 to 60 satellites). These numbers indicate that information saturates beyond 48 satellites.

Figure 12: Time series of the SDs of the hourly differences between the ECMWF IWV field and the ML mapped IWV fields, for the different satellite constellations.

To further study how the error from the first network propagates into the output of the second network, we also mapped the 'true' ECMWF IWVs spatially, for the same 50% dataset as the mappable-IWVs. Then, we compared the ECMWF IWV fields, and the ones mapped with the trained NN models, i.e., computed their differences. For 12, 24, 36, 48, and 60 satellites the average SDs are 2.6, 2.1, 1.8, 1.6, and 1.5 mm; an increment of 12 satellites improves the results by  $\sim$ 19% (12 to 24 satellites),  $\sim$ 14% (24 to 36 satellites),  $\sim$ 11% (36 to 48 satellites), and  $\sim$ 6% (48 to 60 satellites). The average MREs are 9.3%, 7.4%, 6.3%, 5.7%, and 5.3.%; an increment of 12 satellites improves the results by  $\sim$ 20% (12 to 24 satellites),  $\sim$ 15% (24 to 36 satellites),  $\sim$ 10% (36 to 48 satellites), and  $\sim$ 7% (48 to 60 satellites).

The statistics of the mappable-IWVs, compared to ECMWF-based IWVs (with the same spatial distribution), are about 20% worse in terms of SD and about 25% worse in terms of MRE. This is the additional error in the final IWV fields that propagates from the 1<sup>st</sup> NN to the 2<sup>nd</sup> NN. We conclude that the reconstruction of an AR structure requires a constellation of at least 48 satellites.

# 4. Results based on available RO refractivity observations

We investigate how well the COSMIC-2 mission, the most scientific RO-based mission, performs to reconstruct the AR of our scenario, shown in section 2.1. We use the COSMIC-2 data available on the website (https://registry.opendata.aws/gnss-ro-opendata) on the Registry of Open Data on AWS (Leroy et al., 2024), where RO observations from different missions are collected. We evaluate the refractivity at 2 km, a height where small-scale structures related to water vapor are visible and so also the AR. To have a reasonable amount of data, we use observations for six days, from October 22<sup>nd</sup> until October 27<sup>th</sup>; i.e., we include two days before and after the AR. This resulted in approximately 31,000 observations globally from the COSMIC-2 mission (UCAR, 2025).

For our scenario, other observations were available from different scientific missions such as MetOp (EUMETSAT, 2025), Paz (https://paz.ice.csic.es/, 2025), KompSat5 (eoPortal, 2024), TerraSAR-X (GFZ, 2025), as well as the publicly available

commercial data from Spire (https://spire.com/blog/maritime/radio-occultation-in-less-than-500-words/, 2021). Adding these observations results in approximately 54,000 global observations of refractivity at 2 km geopotential height. We performed two experiments; one where only COSMIC-2 data were used, and a second one where all the available data are used. In this way we demonstrate the importance of higher density observations for monitoring and resolving ARs.

Using only COSMIC-2 data, we had 673 occultations for  $24^{th}$  and  $25^{th}$  of October in the AR region; this results in ~14 occultations hourly. Using data from COSMIC-2, MetOp, Paz, KompSat5, TerraSAR-X, and Spire, we had 1173 occultations for  $24^{th}$  and  $25^{th}$  of October in the AR region; this results in ~24.4 occultations hourly. The global datasets are visualized in *Figure 13*, and a refinement in the AR region is shown in *Figure 14*. To fill hourly every 1° latitude-longitude bin of the AR region that we selected we would need 3,500 occultations. As we can see in Figure 13 and Figure 14, the COSMIC-2 mission has observations up to ~45° latitude, while observations from all the missions fully cover the investigated region.

Figure 13: Refractivity at 2 km height from COSMIC-2 (left panel) and all available missions (right panel), during 22<sup>nd</sup> and 27<sup>th</sup> October 2021. The purple rectangle highlights the AR scene.

Figure 14: Refractivity at 2 km height from COSMIC-2 (left panel) and all available missions (right panel), during 22<sup>nd</sup> and 27<sup>th</sup> October 2021. This is a zoom of Figure 13 that highlights the AR scene.

We initially perform ML experiments where we split the data into 80% training (and validation) dataset and 20% testing dataset. We see an improvement of about 15% in standard deviation (SD) of the residuals of the test datasets when the data

from all the missions are used compared to the case when only COSMIC-2 data are used. We produce hourly maps of refractivity at 2 km using the COMSIC-2 dataset and the dataset including all missions. *Figure 15* and *Figure 16* display the ML-mapped refractivity fields (center and right panels), and the ECMWF 12-h forecast maps (left panels), for two epochs during the AR occurrence. We computed the hourly SDs of the differences between ECMWF and ML-based maps. Using all RO missions rather than just COSMIC-2 reduces the SD from 11.49 *N*-units to 9.42 *N*-units over the entire timespan, an approximately 18% improvement. Considering only the latitudes covered by COSMIC-2, then the SD falls from 11.40 *N*-units to 10.59 *N*-units when considering all missions, an approximately 7% improvement. From *Figure 15* and *Figure 16* we can see that the greater number of observations better tracks the AR shape for a longer duration; at 15:00, on October 25<sup>th</sup> (*Figure 16*) a narrow blue stream when we use all the observations (right panel) is visible, while using only COSMIC-2 observations results in dry values, thus, losing the AR shape (center panel).

Figure 15: Refractivity fields at 2 km height, for 24 October 2021, at 06:00 UTC. The left panel shows the ECMWF forecast field; the center panel the ML-mapped field when using COSMIC-2 observations as input to train the NN; and the right panel the ML-mapped field when using observations from all the available missions as input to train the NN.

Figure 16: Refractivity fields at 2 km height, for 25 October 2021, at 15:00. The left panel shows the ECMWF forecast field; the center panel the ML-mapped field when using COSMIC-2 observations as input to train the NN; and the right panel the ML-mapped field when using observations from all the available missions as input to train the NN.

# 550 5. AR in ground-based GNSS data

We saw that mapping IWVs over land is not as accurate as over ocean. To enhance AR mapping over land we also exploit IWV estimated from ground-based (GB) GNSS. The IWVs from GB GNSS are computed from the zenith total delays, which are a principal output of GNSS processing. The dry delay is modelled using an empirical model, such as the Saastamoinen model (Saastamoinen, 1973), and subtracted from the total delay. The remaining wet delay is converted into IWV using the mean temperature from ERA5 (Yuan et al., 2023). NNs are not needed to produce mappable-IWVs, unlike the case of ROs.

#### 5.1 AR in GB GNSS data

GB GNSS determinations of IWV are elevated when an AR crosses a station, just as GNSS RO determinations of IWV are elevated when they are located directly in an AR. The spatial density of GB GNSS stations is especially useful in improving the mapping of the fine structure of ARs, especially in the case of Pacific-coast stations for ARs in the Pacific basin. Moreover, GNSS GB determinations of IWV can be produced at a very high temporal resolution (down to 5 or 15 minutes). *Figure 17* (right panels) show the IWV from ECMWF 12-h forecast for two epochs, one during the AR life cycle and one when it has ended. The black dots mark the GNSS GB stations. The average GNSS stations distance over the entire scene (displayed here) is 20 km, with much smaller separations on the Pacific Coast where ARs occur. The left panels display the IWVs processed from the Nevada Geodetic Laboratory (NGL) (Blewitt et al., 2018). We can clearly see the higher values when the AR is occurring (*Figure 17*, top panels), and much lower IWV values when it has finished (*Figure 17*, bottom panels).

Figure 17 (center panels) displays the IWV from the ECMWF model, interpolated at the NGL stations' locations. Their agreement with the GNSS IWVs (left panels) is visibly very good. To be consistent with the results obtained from the test case of simulated RO, we also use ECMWF to simulate IWVs for the ground-based GNSS network. In Section 5.2, we use the simulated IWVs to display the improvement in AR shape and path over land with GB GNSS compared to RO. Similar figures are produced for the entire scenario and stacked together as an animation, provided as supplementary material to this paper.




Figure 17 AR scenario on October 24<sup>th</sup> at 13:00 when the river is occurring (top panels) and on October 26<sup>th</sup> at 09:00 when the river has finished (bottom panels). Right panels: the ECMWF field, the dots in black are the GB GNSS stations. Left panels: IWVs from GNSS processing, from NGL (Blewitt et al., 2018). Center panels: ECMWF IWVs interpolated at the GB GNSS locations. The green squares are used to highlight the AR location over ground. All IWV values are for precipitable water in mm.

# 5.2 Continuous monitoring of ARs over land using mapped GB GNSS data




We train a NN to spatially map the simulated IWVs for the GNSS network, shown in *Figure 17*. Then, we map IWVs at the ECMWF grid and compute the differences between the "original" ECMWF IWV field and the ML-mapped IWV field. For one epoch during the AR, Figure 18 displays ECMWF IWV field (left panel), the residuals for the case that simulated GB GNSS data were used (right panel), and the residuals for the case that ROs for a constellation of 60 satellites were used to map IWVs following the framework of Section 3.2.3 (center right panel). The results are displayed for the ground area surrounded by GB GNSS stations (30°N to 50N and 90°W to ~125°W); mapping outside this area leads to poor results due to extrapolation. Since the ML model is trained with a dataset collected in a defined area (such as the one shown in *Figure 17*), the model would fail to generalize for new meteorological environment. This is especially the case for a highly variable gas (in space and time) such as water vapor. From Figure 18, we clearly see that the residuals when mapping GB data are much smaller than when using RO data.

Figure 18 The left panel shows the ECMWF forecast IWV over ground, during 25 October 2021 at 04:00. The other panels show the differences between the ECMWF field and the ML mapped fields for the 60-satellites constellation (center right panel), and for the case when using GB GNSS data (right panel).




Figure 19 shows the hourly SDs of the residuals for the different cases; again, these are the statistics over land for the area in the GB GNSS network shown in the Figure 19. There is a clear improvement of about 65% in SD when using GB GNSS data. Additionally, GB GNSS IWV induces no error from adjacent soundings with different IWV as is incurred by RO IWV.

Figure 19 Time series of the SDs of the hourly differences between the ECMWF IWV field and the ML-mapped IWV fields over land, for the different satellite constellations and for the GB GNSS case (black curve).

We also train NNs where we simultaneously use IWVs from GB GNSS and mappable-IWVs from RO. After tuning, the hyperparameters of the NNs when using GB data are: "number of epochs" equal to 15000, "batch size" equal to 4000 and "learning rate" equal to 0.001. The NNs had 5 layers with 512 neurons in the first layer and 128 neurons in the other layers. The results were similar to using only the GB data because the number of observations does not change much when considering ROs over land. For the 4-day scenario, there are ~330K GB observations and ~7.5K RO observations over land (for the 50% test dataset used to generate the mappable-IWVs, see *Figure 9*) assuming a 60-satellite constellation. The simultaneous use of GB IWVs and RO mappable-IWVs would be beneficial in areas with low-density GB networks; our study area has one of the

densest GB networks globally. We conclude that GB GNSS data are extremely helpful to reconstruct continuously and accurately the IWV structure of the selected AR over ground.

#### 615 Summary, conclusions and discussion




In this work, we investigate GNSS RO as a method to reconstruct (and monitor) ARs. The high vertical resolution of GNSS RO is important to capture the amount of water vapor at different altitudes, for RO profiles located in the atmospheric planetary boundary layer. One drawback with GNSS RO is its horizontal and temporal resolution. Indeed, the narrow width of ARs and short timescales associated with ARs make it difficult to use current RO densities to reconstruct ARs. With current RO data, only few (tens) of occultations are available inside an AR scene (here defined  $\sim 70^{\circ} \times 50^{\circ}$  in latitude/longitude). To ameliorate this problem, we leverage our previous work, where we developed an ML approach to map GNSS ROs, showing that our method can increase the resolution of RO observables both spatially (horizontally) and temporally. We focus our analysis on an AR that hit the West Coast of the US during the 24th and 25th of October 2021. This AR led to heavy precipitation, flooding events (and warnings), storms (treefalls), power outages and minor mud slides.

We investigate how different prospective GNSS RO LEO constellation configurations might impact the reconstruction of the AR structure. Using an orbit propagator for the LEO satellites and actual ephemerides of the GNSS satellites, we calculate the geolocations and the times that RO would occur. We considere RO soundings, fore and aft, that would be obtained using the GPS, GLONASS, Galileo, and the BeiDou GNSS constellations. We use NWP forecasts from ECMWF with 0.1° latitude/longitude resolution to interpolate ECMWF refractivities and IWVs to the geolocations and times of the occultations, 630 thus, producing simulated RO observations. By mapping the simulated ROs to the original ECMWF grid, we can perform closed-loop validations of our results.

In this work, we have two main objectives:

- The first goal is to design appropriate LEO constellations for detection of ARs. We consider Walker constellations because of their uniform RO sounding coverage, symmetrical distribution of satellites and scalability. We test constellations consisting of 12, 24, 36, 48 and 60 satellites. The main orbital parameters that we investigate are the number of planes and the inclination. We find the optimal inclination to be polar orbits, which minimize temporal variability in regional sounding numbers. For a 12-satellite constellation, 3 orbital planes and 90° inclination lead to the best results for reconstructing ARs. For 24, 36, 48, and 60 satellites, the best option is using 6 orbital planes with an inclination of 85°.
- After finding the best constellations, and simulating the RO observations based on ECMWF, the second objective is 640 to define the minimum number of satellites that is appropriate to reconstruct accurate and continuous AR fields. Initially, we map the refractivity at 2 km iso-height. In this case, we notice important improvements when increasing the number of satellites from 12 to 24 and from 24 to 60. A constellation of 36 satellites can reconstruct the AR structure well. Then, we map the IWV in the AR scene, using profiles of refractivity. In this case, we use an architecture with two consecutive NNs. The first NN is used to map the refractivity profiles into IWV; the output being mappable-IWVs. We can see that mapping refractivity into

IWV is less accurate (at least in terms of MRE) over land compared to locations over the ocean. The second NN uses the output of the 1<sup>st</sup> NN, i.e., the mappable-IWVs, to produce continuous fields of IWVs, i.e., to map IWV spatially and temporally. In this case, a 48-satellite constellation seems more appropriate to reconstruct the IWV field during the AR. By studying fields of IWV mapped using as inputs (1) the ML-based mappable-IWVs and (2) the original ECMWF-based IWVs, we can also conclude that the error propagating from the 1<sup>st</sup> NN network to the 2<sup>nd</sup> NN is ~20% in terms of SD and ~25% in terms of MRE.

We point out that the ML results in this study are an average of an ensemble of 10 trained NNs.

We use observations of current RO missions and map continuous fields of refractivity at 2 km height to reconstruct the AR path and shape. We map refractivities from the COSMIC-2 mission and refractivities from all RO missions horizontally during the AR occurrence. The hourly RO counts are 14 for COSMIC-2 only and 24.4 for all missions. Due to the low density (in both cases), while we are able to map parts of the AR structure, it is not possible to continuously produce refractivity maps that reconstruct appropriately the AR. However, these tests show that including more RO soundings can benefit the reconstruction of the AR.






To enhance the reconstruction of the AR structure over land we use observations of GB GNSS. IWVs from GB GNSS have been used for a long time to monitor water vapor in the atmosphere. Here, after visualizing the AR sensed by the GB network, we map GB IWVs into gridded products over ground. In this case, we only need one NN to map IWV spatially. The high temporal and spatial distribution of GB GNSS leads to much improved (~65%) AR structure over ground compared to the case when we use RO. The combination of space- and ground-based GNSS observations works very well in areas with high-density of ground-based GNSS observations. Many atmospheric rivers occur in Western Europe, Australia, New Zealand and Chile that have local ground-based networks with a relatively high density.

There is no question that current operational RO retrievals suffer from biases associated with super-refraction. The biases are typically -4% in refractivity, which leads to approximately -25% is water vapor. It is a long recognized and venerable problem. On the other hand, at least three methods have been proposed for mitigating the bias associated with super-refraction: (1) with collocated water vapor radiometer soundings, (2) with collocated infrared or microwave nadir radiance data, or (3) by consideration of the synchronous signal of the RO transmitter off of the ocean surface in maritime environments. None of these algorithms have been exercised on program-of-record data at scale yet, and certainly modifications to GNSS RO receiver design and formation flying of RO satellites with radiance sounder satellites are expected to help in the future. We leave the problem of super-refraction to the RO retrieval science community, which is very actively making progress along these lines.

Our study aims to provide a framework to reconstruct ARs from space- and ground-based GNSS observations. An important part of our study is the demonstration of the ML framework with simulated observations. While the study sets a baseline to map and study IWV fields for AR events, there are some limitations worth mentioning due to our simulation design and focus on this particular AR:

- Distribution of GB GNSS networks: the GB network in the US West Coast of the Pacific is one of the densest GNSS networks. In areas with low density of GNSS stations, for example in Africa, the combination of GB and RO data will bring less improvement. In this case, we would benefit more from simultaneous IWVs from GB and RO over ground.
- IWV at RO locations: the IWV at the RO location will often be underestimated due to missing RO data in the profile for the lowest hundred meters. While this is not a problem with simulated data, there will be an increased uncertainty from training ML models that map IWV from refractivity profiles for real data. A possible solution is to train the RO profiles on other datasets that do not miss the lowest IWV information such as datasets of reprocessed weather models or microwave radiometers.
- Duration, location and lateral movement of ARs: the results we have presented are focused on one AR, a relatively representative AR event. However, the duration, location and movement vary for different ARs, which means that while our results can be generalized to some extent, a similar analysis would be appropriate when considering other ARs.
- Possible combination with Special Sensor Microwave Imager/Sounder (SSMI/S) data: while in this work we focus on spaceand ground-based GNSS data, SSMI/S satellite data are a viable IWV source to combine with RO over ocean. An important
  consideration for SSMI/S is that such datasets may contain large gaps under rainy and cloudy conditions that are typical in
  ARs. In addition, SSMI/S and other microwave radiometers can obtain high sounding densities, but they are poor at
  discerning the vertical structure of water vapor in a column, and they are far more costly instruments. As a consequence,
  they typically do not obtain good coverage of the diurnal cycle and leave significant gores in their sampling patterns at low
  latitudes.
- Horizontal resolution of AR: in our study, in case of simulated observations we assume RO data with very high horizontal resolution. In case of real data, RO resolution is very good in the cross-track direction with 1.5 km resolution. The horizontal resolution of an RO sounding in the along-track direction has never been objectively quantified. It almost certainly depends on the effective vertical resolution of the retrieval, which is detected by radioholographic filters that are applied in "physical optics" retrievals and on any other smoothing of the raw data or retrieved profiles in the retrieval algorithm. In theory, it should be possible to obtain effective vertical resolution of ~100 meters in an RO retrieval. The horizontal path of an RO ray through a 100-meter atmospheric layer is ~70 km. We can consider this an optimistic horizontal resolution of an RO sounding.

Code and data availability. The code associated with this study is available from the corresponding author on reasonable request. The datasets generated during and/or analyzed during the current study are available from the corresponding author on reasonable request. The RO data are freely available on the registry of Open Data on AWS. The ECMWF data are a product of the European Centre for Medium-Range Weather Forecasts (ECMWF) (© ECMWF).

Author contributions. The work presented in this paper was carried out in collaboration between all authors. ES did the preprocessing of the data, implemented the experiments, and performed data analysis. SL helped develop and improve part of

the code to calculate occultation events, provided advice on data analysis and contributed to the interpretation of the results.

KC provided advice on data analysis and contributed to the interpretation of the results. JC did the initial implementation of finding occultation events for GNSS and Walker constellations. BS provided the ECMWF data and helped with the interpretation of the results. ES and SL prepared a draft of the paper, and all authors edited the manuscript.

Competing interests. The authors declare that they have no conflict of interest

Acknowledgements. The authors would like to thank the Swiss National Science Foundation (SNSF) for financing this work (project number P500PN\_222290). S. Leroy was supported by the Decadal Survey Incubator Science program of the U.S. 505 National Aeronautics and Space Administration, grant 80NSSC22K1003.

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
