# Peer review of "A feasibility study to Reconstruct Atmospheric Rivers using spaceand ground-based GNSS observations"

_EGUsphere, 2025_

## Referee Comment (RC1)

**Review of the *AMT* manuscript:**

**A feasibility study to Reconstruct Atmospheric Rivers using space and ground-based GNSS observations**

Endrit Shehaj, Stephen Leroy, Kerri Cahoy, Juliana Chew, Benedikt Soja

This is an interesting simulation study with useful results. The potential impact is, however, limited due to some choices in the design of the study (see my general comments below). The presentation of the results would merit more attention. I have started to compile a list of specific comments and corrections, but this list is not yet exhaustive.

**General comments**:

(1) Study design: You correctly state that "Commercial RO companies around the world are now projecting RO sounding densities exceeding 100,000 daily" but nevertheless you only consider up to 60 satellites in your simulations. Already in 2024 the YunYao constellation alone reached 22 RO satellites and the Tianmu constellation 23 satellites. For a feasibility study it would have been desirable to go beyond what is actually already there.

(2) Study design: "we used refractivity profiles in the geopotential height interval 1–10 km above the ground in 1-km intervals to compute and map IWV". Why?
There is very little water vapor at 9 km or at 10 km. But there is a lot of water vapor in the lowest km (which is critically important, when IWV is computed). Why did you not consider this information?

(3) Study design: All results are based on a single AR event in October 2021, therefore the general conclusions should be considered with some care (e.g., the lateral movement of ARs can be quite different from case to case). The proposed combination of space and ground-based observations may work well in the selected study region, with a high density of ground-based GNSS receivers. In other regions this will not be the case.
Furthermore, most of the AR activity is over the ocean, where essentially no ground-based GNSS receivers are available. Why didn't you consider an alternative information source for IWV values, like SSMI/S satellite data?

(4) The study uses "Perfect observations". While this fine as a first approach, the limitations should at least be discussed (in more detail). E.g., when IWV values are computed from actual RO profiles, there is a problem due to (often) missing data in the lowest few hundred meters of the RO profile.

(5) Please follow the AMT guidelines throughout, e.g. "Kursinski, et al., 1997" (line 42) should be written as "Kursinski et al., 1997" (without ","). Use "Leung and Qian" instead of "Leung & Qian" …
Please update the bibliography:
List all author names, "et al." is not appropriate here.
Please use DOIs throughout.
Please check for missing information throughout, e.g. for "Chen, X. et al., 2018.": issue ? page numbers ?

**Specific comments and technical corrections:**

Line 45: For your study, a specific advantage of the RO method is that it allows to see water vapor below clouds.

Line 46: Since ARs are narrow, you should also discuss the horizontal resolution of RO data.

Line 52: It would be helpful to give approximate numbers for the atmospheric Rossby radius of deformation

Line 60: "ARs can release massive amounts of moisture in the form of precipitation and snowfall" snow is a form of precipitation.

Line 65: "river, (NOAA, 2023)." --> "river (NOAA, 2023)." without ","

Line 66: "(Zhang, et al., 2018) shows .. " --> "Zhang, et al. (2018) show "

Line 69: Clicking on this link produces a 404 error ("Bad request"), because the part after "10.1002/" is not transferred.

Line 72: Also a click on this link produces a 404 error.

Line 75: I would suggest to avoid the acronym NWM for "numerical weather model", but to use NWP model instead.

Line 107: "Present retrieval algorithms for RO fail in the planetary boundary layer" is a bold statement, which is not entirely true. And the formulation "Present retrieval algorithms for RO fail in the planetary boundary layer (PBL) in which ARs evolve due to spherical asymmetry …" is unclear. You don't mean that "ARs evolve in the PBL due to spherical asymmetry", right?

Line 109: It is true that super-refraction induces negative biases in retrieved refractivity – if it is present. However, in contrast to what one might assume, RO data can e.g., often be exploited down to the surface right in the center of ARs, where water vapor values are high but gradients are not.

Line 126: "We focus on the Pacific basin". You just focus on (parts of) the *North* Pacific, right?

Line 134: "at 19:00" Is this local time, or UTC?

Line 142: "the wettest day ever" Could you quantify this, in terms of mm precipitation?

Line 159: "In this section, we our approach" Please add the missing word.

Line 173: you should add "medium Earth orbit" as explanation for "MEO"

Figures 2, 3: "Nb." Is an uncommon abbreviation for "Number".

Figure 3: I would suggest to use at least the same font size as in Fig. 2.

Line 262: "This reflects the terrain altitude shown in Figure 6 (bottom subplots)." I don't see any bottom subplots in Figure 6.

Line 275, equation (2): Do you count the levels from the "top of the atmosphere" downwards, as ECMWF does? Maybe you explain this in a bit more detail. I assume that you used the surface pressure as "$p_{i+1/2}$" for level 137?

Line 276: "gravitational acceleration" In fact this is the "acceleration due to gravity" (note the different meanings of "gravitation" and "gravity").

Line 447: "These numbers indicate that information saturates beyond 48 satellites." This might be true for the IWV field as a whole, bot not when you analyze ARs – which is the main focus of your study. See your own comment: "large residuals seen over the ocean are caused by abrupt horizontal discontinuities in IWV, especially near the AR itself, where large refractivity gradients occur within a few kilometers." (line 395+).

Therefore, also the conclusion "In this case, a 48 satellites constellation is appropriate to reconstruct the AR structure." (Line 607) is misleading, since you did not really analyze the structure of the AR.

---

## Referee Comment (RC2)

Review Comments to the manuscript on **"A feasibility study to Reconstruct Atmospheric Rivers using space and ground-based GNSS observations"** submitted to Journal of AMT.

Overall Comment:

The manuscript attempts to highlight the results of an important feasibility study on reconstruction of atmospheric rivers (ARs), known to be a chief source of bulk of atmospheric water vapour transport from tropics to mid-latitude regions. The study resorts to use of both space-based as well as ground-based GNSS observations within the framework of machine learning NN architectures. The subject of the study is quite significant and the use of ML based exploratory nature of the work is relevant given the surge in the use of ML techniques in recent times. The study is meticulously planned, thoroughly detailed and thoughtfully implemented, even if for a localized region of study with a signature of AR. The craftsmanship with NN architectures is well appreciated, though study with some of the advanced ML approaches, e.g. ensemble learning etc. would have been comparative. Authors are admonished to think of it as future scope of the work. Nevertheless, the study highlights the potential effectiveness of GNSS RO data when used additionally with ground based networks. The manuscript is well-written, organized nicely and representations are clear. The manuscript is recommended for publication after a few suggested modifications as illustrated in "Detailed major comments" and "other minor comments" below.

Detailed major comments:

1. In figure 1, what is the grid resolution of the ECMWF 12 h forecast? It must be mentioned in the text. A similar structure with a long streak is seen in red below the rectangle. What objective basis is used by the authors to identify the AR in figure 1? Justification required. Sub-section 2.1 needs to be strengthened with substantive basis and characterization of AR scenarios in terms of parameters to distinguish them from other features of similar nature.

2. In sub sub-section 3.1.1, authors highlight the results for 65º inclination but the same are not shown in either figures 2 (shows 70º, 80º, 90º and 100º inclinations) or figure 3 (85º inclination). Authors to re-draw the figures 2 and 3 to incorporate the results for 65º inclination also.

Other minor comments:

1. Line 12: Expand the acronym "GNSS" as it ought to be when used for first time. In Line 39: correct practice is to put the acronym in parenthesis and expanded form outside and not the vice versa.
2. Line 46: remove comma after the word "power".
3. Line 48: Put an apostrophe after "GNSS constellations".
4. Line 49: Remove comma after "coverage".
5. Line 50: remove "and".
6. Line 65: Mention the time period used to compute the "average flow…" from the source NOAA, 2023. Remove comma after "river".

7. Line 91: Correct "Similarly" to "Similar".
8. Line 118: Same as suggested in minor comment (1) but for ECMWF.
9. Line 159, 160: Correct the sentence. Put apostrophe after the word "constellations".
10. Line 166: Once defined, use the acronym "AR" everywhere.
11. Line 196: AR region longitude range to be corrected in conformity with figure 1.
12. Line 197, 198: remove comma after "85°" and after "planes".
13. Line 297: correct the syntactic error in the sentence "This makes … related caused ...". Also remove the comma after "implementation".
14. Line 329: add the clause "in the given order" after "5 constellations".

---

## Author Comment (AC1)

**Reviewer #2:**

**Overall Comment:**

**The manuscript attempts to highlight the results of an important feasibility study on reconstruction of atmospheric rivers (ARs), known to be a chief source of bulk of atmospheric water vapour transport from tropics to mid-latitude regions. The study resorts to use of both space-based as well as ground-based GNSS observations within the framework of machine learning NN architectures. The subject of the study is quite significant and the use of ML based exploratory nature of the work is relevant given the surge in the use of ML techniques in recent times. The study is meticulously planned, thoroughly detailed and thoughtfully implemented, even if for a localized region of study with a signature of AR. The craftsmanship with NN architectures is well appreciated, though study with some of the advanced ML approaches, e.g. ensemble learning etc. would have been comparative. Authors are admonished to think of it as future scope of the work. Nevertheless, the study highlights the potential effectiveness of GNSS RO data when used additionally with ground based networks. The manuscript is well-written, organized nicely and representations are clear. The manuscript is recommended for publication after a few suggested modifications as illustrated in "Detailed major comments" and "other minor comments" in the attached pdf.**

Dear reviewer, thank you very much for taking the time to review our work. We really appreciate your comments and your positivity towards our work.

Before answering to each specific comment, I wanted to point out that our results are an output of an ensemble of 10 different NNs (each with slightly different results due to random initialization of model parameters, stochastic optimization algorithms that randomly sample the data points or possible GPU precision and optimization implementation). This has not been strongly highlighted in the manuscript, only mentioned in lines 296-299. To further emphasize this, the following sentence has been added in the conclusions:

'We point out that the ML results in this study are an average of an ensemble of 10 trained NNs.'

Below, you can find our answers to all your points in an extended version.

**Detailed major comments:**

**1. In figure 1, what is the grid resolution of the ECMWF 12 h forecast? It must be mentioned in the text. A similar structure with a long streak is seen in red below the rectangle. What objective basis is used by the authors to identify the AR in figure 1? Justification required. Sub-section 2.1 needs to be strengthened with substantive basis and characterization of AR scenarios in terms of parameters to distinguish them from other features of similar nature.**

Thank you for pointing this out. The ECMWF data that we have used in this work are at 0.1° resolution. This is mentioned later in the text in section 3.1.2 and at the conclusions section. As suggested, I also added this in the Figure caption as follows:
'*AR scenario visualized using 12-h forecast ECMWF data, with a resolution of 0.1°.*'.

Regarding the AR identification, we do not identify it ourselves but rather use an AR that has been identified/reported by NOAA:
https://www.weather.gov/mtr/AtmosphericRiver_10_24-25_2021#:~:text=Atmospheric%20River%20October%2024%2D25%2C%202021&text=Higher%20elevations%20of%20the%20North,the%20risk%20of%20fire%20season.

To further specify this, we have changed the following sentence in section 2.1:
'*Figure 1* depicts the AR scenario that we have identified on the website of US National Weather Service, (NOAA, 2021).'. into '*Figure 1* depicts the AR scenario reported on the website of US National Weather Service, (NOAA, 2021).'

We would like to clarify that the red structure represents very dry area where the refractivity values are the lowest (dark red in the colorbar), and the AR is the area with highest moisture, i.e. highest refractivity values (dark blue).

In the introduction, we have mentioned the main characteristics of AR as follows:

'*ARs are narrow maritime atmospheric low-level jets that transport large amounts of moisture from the Tropics into the mid- and high latitudes, often impinging on the continents (Newell, et al., 1992), (Zhu & Newell, 1994), (Newell & Zhu, 1994). ARs can release massive amounts of moisture in the form of precipitation (NOAA, 2023). Depending on the size and intensity of the AR, it might lead to extreme precipitation …*'

and

'*AR widths are typically 1000 km, and their lengths 2000 km or longer (Zhu & Newell, 1998). The total precipitable (column-integrated) water vapor is at least 20 mm (20 kg m-2) (Ralph, et al., 2004). While they cover only 10% of Earth's circumference, ARs are still responsible for 90% of the total meridional moisture transport (Zhu & Newell, 1998).*'

To further distinguish the AR, we refer to our supplementary material, where videos of ECMWF data are also reported (in *Video_IWV_maps.mp4* and *Video_Refractivity_maps.mp4*). The AR is a more persistent structure on the 24th and 25th of October that fits the AR characteristics mentioned in the introduction.

We have also added the following sentence:
'The evolution of the selected AR can be seen in the supplementary material where videos of ECMWF refractivity and IWV fields are included.'

**2. In sub sub-section 3.1.1, authors highlight the results for 65o inclination but the same are not shown in either figures 2 (shows 70o , 80o , 90o and 100o inclinations) or figure 3 (85o inclination). Authors to re-draw the figures 2 and 3 to incorporate the results for 65o inclination also.**

Thank you for pointing this out. In the following figure we show the results for all evaluated inclinations:

[Figure]

*Fig: Hourly number of RO counts for Walker constellations of RO receivers in orbit planes with inclinations of 65°, 80°, 90°, and 100° in the AR region. On the left are displayed the cases for 12-satellite constellations and on the right the cases for 60-satellite constellations. The top, middle and bottom panels represent the cases of 3, 4 and 6 orbital planes.*

Since in this figure it is very difficult to distinguish the different curves, we opted to show the results of only some of the inclinations and decided 70, 80, 90 and 100 to keep a similar step. The detailed statistics are then reported for each inclination in Table 1.

We have changed Figure 2 so that the results for 65° inclination are included. To keep the figure as clear as possible, we are showing the results of 65° instead of the results of 70° inclination:

[Figure]

*Figure 2: Hourly number of RO counts for Walker constellations of RO receivers in orbit planes with inclinations of 65°, 80°, 90°, and 100° in the AR region. On the left are displayed the cases for 12-satellite constellations and on the right the cases for 60-satellite constellations. The top, middle and bottom panels represent the cases of 3, 4 and 6 orbital planes.*

The following figure is the equivalent of Figure 3 for 65° inclination:

[Figure]

*Fig: Hourly number of RO counts for 65°-inclination orbit planes, in the AR region, for 12, 24, 36, 48, and 60 satellites. The top, middle and bottom panels represent the cases of 3, 4 and 6 orbital planes.*

We would appreciate the understanding of the reviewer to not add this figure in the manuscript because the manuscript is already very long (~30 pages). The results for 12 and 60 satellites are reported in Table 1 and the readers will be able to see this figure in the discussion section of the manuscript.

**Other minor comments:**

**1. Line 12: Expand the acronym "GNSS" as it ought to be when used for first time. In Line 39: correct practice is to put the acronym in parenthesis and expanded form outside and not the vice versa.**

**2. Line 46: remove comma after the word "power".**

**3. Line 48: Put an apostrophe after "GNSS constellations".**

**4. Line 49: Remove comma after "coverage".**

**5. Line 50: remove "and".**

**6. Line 65: Mention the time period used to compute the "average flow…" from the source NOAA, 2023. Remove comma after "river".**

**7. Line 91: Correct "Similarly" to "Similar".**

**8. Line 118: Same as suggested in minor comment (1) but for ECMWF.**

**9. Line 159, 160: Correct the sentence. Put apostrophe after the word "constellations".**

**10. Line 166: Once defined, use the acronym "AR" everywhere.**

**11. Line 196: AR region longitude range to be corrected in conformity with figure 1.**

**12. Line 197, 198: remove comma after "85° " and after "planes".**

**13. Line 297: correct the syntactic error in the sentence "This makes … related caused ...". Also remove the comma after "implementation".**

**14. Line 329: add the clause "in the given order" after "5 constellations".**

Dear reviewer, thank you very much for reading in detail the manuscript and providing all these corrections. They have been implemented accordingly in the new version of the manuscript.

---

## Author Comment (AC2)

**Reviewer #1:**

**This is an interesting simulation study with useful results. The potential impact is, however, limited due to some choices in the design of the study (see my general comments below). The presentation of the results would merit more attention. I have started to compile a list of specific comments and corrections, but this list is not yet exhaustive.**

Dear reviewer, thank you very much for your thorough review of our work. We appreciate your comments and below, you can find our answers to all your points in an extended version.

General comments:

**(1) Study design: You correctly state that "Commercial RO companies around the world are now projecting RO sounding densities exceeding 100,000 daily" but nevertheless you only consider up to 60 satellites in your simulations. Already in 2024 the YunYao constellation alone reached 22 RO satellites and the Tianmu constellation 23 satellites. For a feasibility study it would have been desirable to go beyond what is actually already there.**

Thank you for pointing this out.

The relationship between number of RO satellites and daily RO sounding counts is not uniform across RO missions. We have simulated RO sounding distributions that consider all the transmitters of the four GNSS, restricted by view azimuth, and assuming no loss of RO data. As a consequence, our 60-satellite constellation obtains 180K RO soundings daily, far in excess of what YunYao, Tianmu, and all other existing RO satellites are obtaining. Finally, the atmospheric science community is barred from publishing in peer-reviewed journals with any data that is not freely available to the global public. Unfortunately, most commercial RO data are not freely available, including YunYao and Tianmu RO data.

In Table 2 of our work you may see that the number of global soundings is 434K with 36 satellites and 724K with 60 satellites for the 4 days of the selected scenario. Therefore, in one day there are more than 100K RO counts with 36 satellites, and ~180K with 60 satellites. In this simulation, we have considered constellations that can collect observations of GPS, GLONASS, GALILEO and BEIDOU, while in the current LEO RO constellations we have many missions that do not provide data from all four GNSS constellations, which can be due to receiver design differences and limitations, such as supporting limited frequencies as well as limits to the number of concurrent channels supported per receiver.

Another factor is that since we are doing simulations, we do not have 'bad observations', and the rate of observations with poor quality or missing data are often either not provided or difficult to obtain statistics on in case of real missions.

Finally, our goal was to design an appropriate constellation to reconstruct IWV maps in an AR scenario. Since we did not see further improvement with further increasing the number of satellites, we decided to limit our simulation at 60 satellites. Indeed, with 48 satellites the IWV is already reconstructed quite well from our results. Consider, for example, Figure 12, where the standard deviation time series are quite similar for 48 and 60 satellites.

To benefit from this comment and improve the paper to further clarify this, we added the following paragraph at the end of Section 3.1.1:

'From our simulations, we achieve about 180K occultations daily with a constellation of 60 satellites. While the number of current LEO RO satellites is already around 60 satellites, the number of RO counts reported is much lower compared to our simulation. This is because of different factors, for example, some LEO RO constellations do not collect data from all available GNSS, in real constellations some data are not reported or difficult to obtain statistics on because they are considered 'bad' or poor-quality observations. In addition, different LEO RO satellites only collect data up to a certain latitude, e.g., COSMIC-2 only collects data from 45°S to 45°N. Finally, as we will see in the following sections, the goal is to simulate a high-performing constellation in order to reconstruct the IWV field in the AR scene; we find that the statistical improvement is relatively small when we consider 60 satellites compared to 48 satellites (see Section 3.2.2 and Section 3.2.3.2), so we did not expand the constellation beyond 60 in this work.'.

**(2) Study design: "we used refractivity profiles in the geopotential height interval 1–10 km above the ground in 1-km intervals to compute and map IWV". Why?**

**There is very little water vapor at 9 km or at 10 km. But there is a lot of water vapor in the lowest km (which is critically important, when IWV is computed). Why did you not consider this information?**

Thank you for pointing this out.

We use refractivity at 1-km intervals up to 10 km above the surface for several reasons. The decision to use 1-10 km with 1-km intervals is a relatively conservative choice, which considers that refractivity in the lowest part of the atmosphere can be less accurate due to ducting and super-refraction, multipath, SNR attenuation or spherical symmetry in the atmosphere, which have less accuracy for strong horizontal gradients. There is no simple retrieval of water vapor from refractivity values in the lower atmosphere without a prior, such as an atmospheric forecast, thus preventing retrieval and vertical integration of water vapor to obtain a column-integrated value. Thus, we rely instead on a few indicative values in the PBL and information on the atmospheric dynamical state, which information is contained in values throughout the tropospheric column. A good machine learning algorithm should be able to discern the meteorology of the local environment and draw on this information to establish a relationship between the refractivity values in the lower troposphere, where water vapor contributes most to refractivity, and column-integrated water vapor. Our motivation is justified by our results, which are found in Table 4.

Also, in the machine learning formulation, we want to use similar inputs for all RO profiles. If we adjusted the intervals and tried to consider height at 0.5 km, for example, we would need to have observations at 0.5 km for all profiles, otherwise we would need to discard the profiles that do not have all intervals available, which would have limited the analysis. So, our choice to use 1-10 km is a compromise for these reasons.

To benefit from this comment and help improve the paper to further clarify this point, we have added the following explanation in section 3.2.3.1:

'We use refractivity at 1-km intervals up to 10 km above the surface for several reasons. The choice of profiles in the 1-10 km interval is conservative considering that in the lowest part of the atmosphere RO-retrieved refractivity is less accurate due to ducting and super-refraction, multipath, SNR attenuation or spherical symmetry in the atmosphere which have less accuracy for strong horizontal gradients. There is no simple retrieval of water vapor from refractivity values in the lower atmosphere without a prior, such as an atmospheric forecast, thus preventing retrieval and vertical integration of water vapor to obtain a column-integrated value. Thus, we rely instead on a few indicative values in the PBL and information on the atmospheric dynamical state, which information is contained in values throughout the tropospheric column. A good machine learning algorithm should be able to discern the meteorology of the local environment and draw on this information to establish a relationship between the refractivity values in the lower troposphere, where water vapor contributes most to refractivity, and column-integrated water vapor. Our motivation is justified by our results, which are found in Table 4.'.

**(3) Study design: All results are based on a single AR event in October 2021, therefore the general conclusions should be considered with some care (e.g., the lateral movement of ARs can be quite different from case to case). The proposed combination of space and groundbased observations may work well in the selected study region, with a high density of groundbased GNSS receivers. In other regions this will not be the case.**

Thank you for the comment. We also analyzed the number of RO soundings for another AR in the UK region. We considered a similar area size and similar scenario duration as the AR in the manuscript. The UK AR occurred between September 30[th] and October 3[rd], 2020, and the area of investigation for this scenario is 20°S-70°N and 50°W-20°E. We achieved the following results, for this AR event, reported similarly to Table 2 in the manuscript:

| Number of LEO satellites | Planes | Inclination [°] | Number of global soundings [K] | Number of soundings in the AR region |
|---|---|---|---|---|
| 12 | 6 | 100 | 139 | 8746 |
| 24 | 6 | 100 | 278 | 17499 |
| 36 | 6 | 100 | 418 | 26260 |
| 48 | 6 | 100 | 557 | 35052 |
| 60 | 6 | 100 | 696 | 43833 |

The number of hourly RO soundings in the AR region is similar to the case reported in the paper. The lower values in the number of global soundings happens because of different TLEs for this event in 2020; in the UK case there are six fewer GNSS satellites compared with the event in 2021. When we use the same TLEs as in our scenario, we achieve similar results for the two cases also in terms of inclination. It is also worth mentioning that for 6 planes, the variation of hourly RO counts is similar for the different inclinations. The variations for different inclinations for 3 and 4 planes are much more noticeable, as reported in Table 1.

To improve the paper based on this comment, we have added the following at the end of section 3.1.1:

'We also studied constellation design for another AR in the UK region between September 30th and October 3rd. The hourly number of ROs and the number of planes is similar to the event on the US West Coast. Differences appear for the inclination, where for the UK event an inclination of 100° appears more appropriate. This difference is mainly caused by the different GNSS TLEs for the two events. Using the same TLEs leads to similar results for both scenarios. We point out that for 6 planes, the variation of hourly RO counts is similar for the different inclinations. The variations for different inclinations for 3 and 4 planes are much more noticeable, as also reported in Table 1.'.

At the end of section 5.2 we mention the following: '*The simultaneous use of GB IWVs and RO mappable-IWVs would be beneficial in areas with low-density GB networks; our study area has one of the densest GB networks globally.*'.

In addition, at the end of the conclusions, we also added the following, based on the distribution of GNSS networks from the Nevada Geodetic Laboratory (https://geodesy.unr.edu/NGLStationPages/gpsnetmap/GPSNetMap.html):

'The combination of space- and ground-based GNSS observations works very well in areas with high-density of ground-based GNSS observations. Many atmospheric rivers occur in Western Europe, Australia, New Zealand and Chile that have local ground-based networks with a relatively high density.'.

**Furthermore, most of the AR activity is over the ocean, where essentially no ground-based GNSS receivers are available. Why didn't you consider an alternative information source for IWV values, like SSMI/S satellite data?**

Our goal was two-fold: (1) investigate whether current RO sounding densities are sufficient to analyze the temporal-spatial morphology of ARs, and, if not, (2) determine how many RO satellites would be needed (in the future) to perform such research. RO has its advantages, which include astonishing vertical resolution, but one of its drawbacks is relatively sparse sounding density compared to passive radiance sounders such as SSMI/S. On the other hand, SSMI/S and other microwave radiometers can obtain high sounding densities, but they are poor at discerning the vertical structure of water vapor in a column, and they are far more costly instruments. As a consequence of the third point, they typically do not obtain good coverage of the diurnal cycle and leave significant gores in their sampling patterns at low latitudes.

The combination of ground-based and RO GNSS data is an intuitive approach since they both provide GNSS observations of the troposphere under all weather conditions. SSMI/S can have large data gaps because of cloudy and rainy conditions, which are common in ARs. We added the following at the end of the conclusions:

'While in this work we focus on space- and ground-based GNSS data, Special Sensor Microwave Imager/Sounder (SSMI/S) satellite data are a viable IWV source to combine with RO over ocean. An important consideration for SSMI/S is that such datasets may contain large gaps under rainy and cloudy conditions that are typical in ARs. In addition, SSMI/S and other microwave radiometers can obtain high sounding densities, but they are poor at discerning the vertical structure of water vapor in a column, and they are far more costly instruments. As a consequence, they typically do not obtain good coverage of the diurnal cycle and leave significant gores in their sampling patterns at low latitudes.'.

**(4) The study uses "Perfect observations". While this fine as a first approach, the limitations should at least be discussed (in more detail). E.g., when IWV values are computed from actual RO profiles, there is a problem due to (often) missing data in the lowest few hundred meters of the RO profile.**

Thank you for the suggestion. At the end of the conclusions section we have added the following discussion:

'There is no question that current operational RO retrievals suffer from biases associated with super-refraction. The biases are typically -4% in refractivity, which leads to approximately -25% is water vapor. It is a long recognized and venerable problem. On the other hand, at least three methods have been proposed for mitigating the bias associated with super-refraction: (1) with collocated water vapor radiometer soundings, (2) with collocated infrared or microwave nadir radiance data, or (3) by consideration of the synchronous signal of the RO transmitter off of the ocean surface in maritime environments. None of these algorithms have been exercised on program-of-record data at scale yet, and certainly modifications to GNSS RO receiver design and formation flying of RO satellites with radiance sounder satellites are expected to help in the future. We leave the problem of super-refraction to the RO retrieval science community, which is very actively making progress along these lines.

Our study aims to provide a framework to reconstruct ARs from space- and ground-based GNSS observations. An important part of our study is the demonstration of the ML framework with simulated observations. While the study sets a baseline to map and study IWV fields for AR events, there are some limitations worth mentioning due to our simulation design and focus on this particular AR.

Distribution of GB GNSS networks: the GB network in the US West Coast of the Pacific is one of the densest GNSS networks. In areas with low density of GNSS stations, for example in Africa, the combination of GB and

RO data will bring less improvement. In this case, we would benefit more from simultaneous IWVs from GB and RO over ground.

IWV at RO locations: the IWV at the RO location will often be underestimated due to missing RO data in the profile for the lowest hundred meters. While this is not a problem with simulated data, there will be an increased uncertainty from training ML models that map IWV from refractivity profiles for real data. A possible solution is to train the RO profiles on other datasets that do not miss the lowest IWV information such as datasets of reprocessed weather models or microwave radiometers.

Duration, location and lateral movement of ARs: the results we have presented are focused on one AR, a relatively representative AR event. However, the duration, location and movement vary for different ARs, which means that while our results can be generalized to some extent, a similar analysis would be appropriate when considering other ARs.

Possible combination with Special Sensor Microwave Imager/Sounder (SSMI/S) data: while in this work we focus on space- and ground-based GNSS data, SSMI/S satellite data are a viable IWV source to combine with RO over ocean. An important consideration for SSMI/S is that such datasets may contain large gaps under rainy and cloudy conditions that are typical in ARs. In addition, SSMI/S and other microwave radiometers can obtain high sounding densities, but they are poor at discerning the vertical structure of water vapor in a column, and they are far more costly instruments. As a consequence, they typically do not obtain good coverage of the diurnal cycle and leave significant gores in their sampling patterns at low latitudes.

Horizontal resolution of AR: in our study, in case of simulated observations we assume RO data with very high horizontal resolution. In case of real data, RO resolution is very good in the cross-track direction with 1.5 km resolution. The horizontal resolution of an RO sounding in the along-track direction has never been objectively quantified. It almost certainly depends on the effective vertical resolution of the retrieval, which is detected by radioholographic filters that are applied in "physical optics" retrievals and on any other smoothing of the raw data or retrieved profiles in the retrieval algorithm. In theory, it should be possible to obtain effective vertical resolution of ~100 meters in an RO retrieval. The horizontal path of an RO ray through a 100-meter atmospheric layer is ~70 km. We can consider this an optimistic horizontal resolution of an RO sounding.

**(5) Please follow the AMT guidelines throughout, e.g. "Kursinski, et al., 1997" (line 42) should be written as "Kursinski et al., 1997" (without ","). Use "Leung and Qian" instead of "Leung & Qian" …**

**Please update the bibliography:**

**List all author names, "et al." is not appropriate here.**

**Please use DOIs throughout.**

**Please check for missing information throughout, e.g. for "Chen, X. et al., 2018.": issue ? page numbers ?**

Thank you very much for checking the references in detail. We have accordingly updated the citations in the text and the bibliography.

**Specific comments and technical corrections:**

**Line 45: For your study, a specific advantage of the RO method is that it allows to see water vapor below clouds.**

Thank you for noticing. This is what we mean when we say all weather capabilities. We have modified this part as follows:

'…, all weather capabilities (not affected by clouds and rainfall), …'.

**Line 46: Since ARs are narrow, you should also discuss the horizontal resolution of RO data.**

Thank you for pointing this out. We added: 'The horizontal resolution of RO is 1.5 km in the cross-track direction. The horizontal resolution of an RO sounding in the along-track direction almost certainly depends on the effective vertical resolution of the retrieval. Through a 100-meter atmospheric layer, the horizontal path of an RO ray is ~70 km. This can be considered an optimistic horizontal resolution of an RO sounding.'.

We also added at the end of the conclusions the following:

'Horizontal resolution of AR: in our study, in case of simulated observations we assume RO data with very high horizontal resolution. In case of real data, RO resolution is very good in the cross-track direction with 1.5 km resolution. The horizontal resolution of an RO sounding in the along-track direction has never been objectively quantified. It almost certainly depends on the effective vertical resolution of the retrieval, which is detected by radioholographic filters that are applied in "physical optics" retrievals and on any other smoothing of the raw data or retrieved profiles in the retrieval algorithm. In theory, it should be possible to obtain effective vertical resolution

of ~100 meters in an RO retrieval. The horizontal path of an RO ray through a 100-meter atmospheric layer is ~70 km. We can consider this an optimistic horizontal resolution of an RO sounding.'.

**Line 52: It would be helpful to give approximate numbers for the atmospheric Rossby radius of deformation**

We added the following: '… Rossby radius of deformation (about 1000 km) and …'.

**Line 60: "ARs can release massive amounts of moisture in the form of precipitation and snowfall" snow is a form of precipitation.**

Thank you for pointing this out, we removed '*and snowfall*'.

**Line 65: "river, (NOAA, 2023)." --> "river (NOAA, 2023)." without ","**

This has been corrected.

**Line 66: "(Zhang, et al., 2018) shows .. " --> "Zhang, et al. (2018) show "**

This has been corrected.

**Line 69: Clicking on this link produces a 404 error ("Bad request"), because the part after "10.1002/" is not transferred.**

Thank you for pointing this. When we copy the following link (https://agupubs.onlinelibrary.wiley.com/doi/toc/10.1002/(ISSN)1944-8007.ATMOS_RIVERS1?page=1) and paste it to the browser it works. It also works when we download the PDF and click on it. It does not work when we open the preprint on the browser and click there.

For now, we have removed the link and instead added the website as a reference as follows:

'The collection 'Atmospheric Rivers', a first effort containing selected research associated to ARs in *Geophysical Research Letters* (agupubs, 2019), encompasses …'.

**Line 72: Also a click on this link produces a 404 error.**

Same as in the previous comment, the website has been added as a reference:

'The more recent collection named 'Atmospheric Rivers: Intersection of Weather and Climate' in *Journal of Geophysical Research Atmospheres* (agupubs, 2024), presents further…'.

**Line 75: I would suggest to avoid the acronym NWM for "numerical weather model", but to use NWP model instead.**

This has been changed.

**Line 107: "Present retrieval algorithms for RO fail in the planetary boundary layer" is a bold statement, which is not entirely true. And the formulation "Present retrieval algorithms for RO fail in the planetary boundary layer (PBL) in which ARs evolve due to spherical asymmetry …" is unclear. You don't mean that "ARs evolve in the PBL due to spherical asymmetry", right?**

Thank you for noticing this. It was a mistake. The paragraph has been corrected (and extended) as follows:

'The RO signal dynamics become very complicated, and consequently tracking an RO signal in the planetary boundary layer (PBL) becomes difficult. Consequently, sometimes RO signals are not able to penetrate deeply into the PBL. In addition, present retrieval algorithms for RO must deal with several complications in the PBL such as spherical asymmetry (Ahmad & Tyler, 1998), (Ahmad & Tyler, 1999), RO signal loss, and truncation by operational retrieval systems. Also, RO retrievals are less precise due to apparent noisy behavior in the retrieved profiles of refractivity and water vapor.'.

**Line 109: It is true that super-refraction induces negative biases in retrieved refractivity – if it is present. However, in contrast to what one might assume, RO data can e.g., often be exploited down to the surface right in the center of ARs, where water vapor values are high but gradients are not.**

Thank you for this context. We added the following sentence at the end of the paragraph:

'In addition, it is important to point out that often RO data can be exploited down to the surface right in center of the ARs, where WV values are high, but gradients are not.'.

**Line 126: "We focus on the Pacific basin". You just focus on (parts of) the \*North\* Pacific, right?**

Thank you for noticing this. It has been corrected to 'We focus on the North Pacific and …'.

**Line 134: "at 19:00" Is this local time, or UTC?**

Yes, the time is UTC. This was clarified as 'at 19:00 UTC'.

**Line 142: "the wettest day ever" Could you quantify this, in terms of mm precipitation?**

The following table named 'Heavy Rain' is reported in the NOAA website of this AR event: https://www.weather.gov/mtr/AtmosphericRiver_10_24-25_2021#:~:text=Atmospheric%20River%20October%2024%2D25%2C%202021&text=Higher%20elevations%20of%20the%20North,the%20risk%20of%20fire%20season.

```
...RECORD DAILY PRECIPITATION RECORDS SET ON OCTOBER 24 2021...

LOCATION               PRECIP(IN)      PREVIOUS RECORD
* * *
SANTA ROSA             7.83*           4.67 IN 1962
KENTFIELD              11.09           6.14 IN 2009
NAPA                   5.35            4.66 IN 1962
SAN FRANCISCO DOWNTOWN 4.02            2.48 IN 2009
SFO                    4.02            2.64 IN 2009
OAKLAND DOWNTOWN       4.28            3.87 IN 2009
```

In the manuscript, we added '… was the wettest day ever for many cities around the San Francisco Bay Area, as reported in table 'Heavy Rain' in (NOAA, 2021)'.

**Line 159: "In this section, we our approach" Please add the missing word.**

'In this section, we present our approach …'.

**Line 173: you should add "medium Earth orbit" as explanation for "MEO"**

This has been added.

**Figures 2, 3: "Nb." Is an uncommon abbreviation for "Number".**

Thank you for noticing. This has been changed to 'No.'.

**Figure 3: I would suggest to use at least the same font size as in Fig. 2.**

Thank you, this was corrected.

**Line 262: "This reflects the terrain altitude shown in Figure 6 (bottom subplots)." I don't see any bottom subplots in Figure 6.**

Thank you for noticing, we had a figure in a previous draft which we removed because it does not add a lot of new information and the manuscript is already relatively long. This was changed to 'This reflects the terrain altitude (not shown here).'.

**Line 275, equation (2): Do you count the levels from the "top of the atmosphere" downwards, as ECMWF does? Maybe you explain this in a bit more detail. I assume that you used the surface pressure as "p_i+1/2" for level 137?**

For this part we have used the information provided on the ECMWF website: https://confluence.ecmwf.int/display/UDOC/L137+model+level+definitions for model levels.

Level 137 is the lowest layer (closer to the ground) and level 1 has the highest altitude.

Please consider the following code provided from ECMWF (see lines 200-243):

https://confluence.ecmwf.int/display/ECC/compute_geopotential_on_ml.py

and lines 191-197 are used to compute pressure at half-levels.

Key atmospheric variables like specific humidity (q), temperature (T), and horizontal winds (u, v) (which are used to calculate IWV) are stored and calculated at the full model levels in the IFS.

Therefore, using the specific humidity values from the full levels, along with the thickness (pressure difference) of the layers (determined by the half levels), we can calculate the total IWV.

**Line 276: "gravitational acceleration" In fact this is the "acceleration due to gravity" (note the different meanings of "gravitation" and "gravity").**

Thank you for noticing this. We point out that in American English 'gravitational acceleration' is acceptable, however we have changed following the reviewer's comment, to avoid confusion for a wider public.

**Line 447: "These numbers indicate that information saturates beyond 48 satellites." This might be true for the IWV field as a whole, but not when you analyze ARs – which is the main focus of your study. See your**

**own comment: "large residuals seen over the ocean are caused by abrupt horizontal discontinuities in IWV, especially near the AR itself, where large refractivity gradients occur within a few kilometers." (line 395+). Therefore, also the conclusion "In this case, a 48 satellites constellation is appropriate to reconstruct the AR structure." (Line 607) is misleading, since you did not really analyze the structure of the AR.**

Thank you for pointing this out.

ML algorithms will always struggle with large discontinuities in IWV, regardless of the number of soundings. It still remains true that the information content on ARs saturates at ~48 satellites.

Considering the sentence **"large residuals seen over the ocean are caused by abrupt horizontal discontinuities in IWV, especially near the AR itself, where large refractivity gradients occur within a few kilometers." (line 395+)**, we would like to point out that the mean relative error is smaller for constellations with more satellites. Then this is also reflected in the $2^{nd}$ NN which does the spatial mapping of the IWV.

In addition, while we do not provide statistics only for the AR shape and path itself, in the supplementary material (the video named Video_IWV_maps.mp4), it is noticeable that fewer satellites lead to the larger residuals not only in the overall field, but also in the AR path itself. This is clear for 12 and 24 satellites. For 36 satellites, the saturation becomes even clearer compared to 48 satellites, especially by the end of the AR ($25^{th}$ of October).

We have refined the statement **"In this case, a 48 satellite constellation is appropriate to reconstruct the AR structure."** as 'In this case, a 48-satellite constellation seems more appropriate to reconstruct the IWV field during the AR event.'.